# A Wnt-mediated transformation of the bone marrow stromal cell identity orchestrates skeletal regeneration

Yuki Matsushita [1], Mizuki Nagata[1], Kenneth M. Kozloff [2], Joshua D. Welch[3], Koji Mizuhashi[1], Nicha Tokavanich[1], Shawn A. Hallett[1], Daniel C. Link[4], Takashi Nagasawa[5], Wanida Ono[1] & Noriaki Ono [1]*

Bone marrow stromal cells (BMSCs) are versatile mesenchymal cell populations underpinning the major functions of the skeleton, a majority of which adjoin sinusoidal blood vessels and express C-X-C motif chemokine ligand 12 (CXCL12). However, how these cells are activated during regeneration and facilitate osteogenesis remains largely unknown. Cell-lineage analysis using *Cxcl12-creER* mice reveals that quiescent Cxcl12-creER[+] perisinusoidal BMSCs differentiate into cortical bone osteoblasts solely during regeneration. A combined single cell RNA-seq analysis demonstrate that these cells convert their identity into a skeletal stem cell-like state in response to injury, associated with upregulation of osteoblast-signature genes and activation of canonical Wnt signaling components along the single-cell trajectory. β-catenin deficiency in these cells indeed causes insufficiency in cortical bone regeneration. Therefore, quiescent Cxcl12-creER[+] BMSCs transform into osteoblast precursor cells in a manner mediated by canonical Wnt signaling, highlighting a unique mechanism by which dormant stromal cells are enlisted for skeletal regeneration.

[1] University of Michigan School of Dentistry, Ann Arbor, MI 48109, USA. [2] Department of Orthopaedic Surgery, University of Michigan, Ann Arbor, MI 48109, USA. [3] Department of Computational Medicine and Bioinformatics, Department of Computer Science and Engineering, University of Michigan, Ann Arbor, MI 48109, USA. [4] Washington University School of Medicine, Division of Oncology, St. Louis, MO 63110, USA. [5] Osaka University School of Medicine, Laboratory of Stem Cell Biology and Developmental Immunology, Graduate School of Frontier Biosciences, Suita, Osaka 565-0871, Japan. *email: noriono@umich.edu

Bone is a highly regenerative organ that maintains its structural integrity and functionality throughout life. Bone marrow stromal cells (BMSCs) are mesenchymal cell populations with versatile functions that provide the support for hematopoietic cells and the source of osteoblasts, underpinning the major functions of the skeleton[1]. A majority of BMSCs assume perivascular locations surrounding sinusoids with a reticular morphology[2] and universally express cytokines such as C–X–C motif chemokine 12 (CXCL12, also known as stromal cell-derived factor 1, SDF1)[3,4] and stem cell factor (SCF, also known as KIT ligand)[5]. Leptin receptor (LepR) is expressed by a wide array of BMSCs that largely overlap with CXCL12-abundant reticular (CAR) cells[6]. These LepR+CXCL12+ BMSCs cells populate the osteoblast compartment in adult mice[7–9], containing a self-renewing population particularly those termed as skeletal stem cells (SSCs) defined by in vitro assays[10]. However, how specific subsets of CXCL12+ BMSCs are activated in response to injury within their native environment and facilitate skeletal regeneration remains largely unknown. In this study, we sought to identify the mechanism underlying the remarkable osteogenic capacity of CXCL12+ BMSCs in vivo under regenerative conditions. To this end, we generated a *Cxcl12-creER* line and performed in vivo lineage-tracing experiments and functional analyses. Our data reveal that quiescent Cxcl12-creER+ BMSCs transform into precursor cells characterized by a SSC-like state in a manner mediated by canonical Wnt signaling during injury responses, and functionally contribute to skeletal regeneration.

## Results

***Cxcl12-creER* marks a quiescent subset of CXCL12+LepR+ BMSCs.** To reveal in vivo cell fates of CXCL12+ BMSCs, we generated a tamoxifen-inducible *Cxcl12-creER* bacterial artificial chromosome (BAC) transgenic line (L289, Fig. 1a). First, we characterized this *Cxcl12-creER* line based on a short-chase protocol. Analysis of *Cxcl12*GFP/+; *Cxcl12-creER*; *R26R*tdTomato mice at postnatal day (P) 28 after a tamoxifen pulse at P21 revealed that *Cxcl12-creER*-marked tdTomato+ (hereafter, Cxcl12CE-tdTomato+) cells overlapped completely with Cxcl12-GFPhigh stromal cells throughout the diaphyseal marrow space (Fig. 1b). These Cxcl12-GFPhighCxcl12CE-tdTomato+ stromal cells assumed a characteristic morphology as reticular cells adjoining endomucin (Emcn)+ sinusoidal endothelial cells (Fig. 1c, Supplementary Fig. 1a) and CD31+ arteriolar endothelial cells (Fig. 1d). In fact, virtually all Cxcl12-GFPhighCxcl12CE-tdTomato+ cells attached Emcn+ sinusoids either through dendrites or by cell bodies (Fig. 1e, 95.7 ± 2.1% of Cxcl12-GFPhighCxcl12CE-tdTomato+ cells, compared to 84.6 ± 6.1% of Cxcl12-GFPhightdTomatoneg cells, *n* = 5). Flow cytometry analysis revealed that Cxcl12CE-tdTomato+ cells represented an extremely small fraction of whole bone marrow cells (0.014 ± 0.002%). *Cxcl12-creER* faithfully marked a subset of Cxcl12-GFPhigh cells upon tamoxifen injection; 27.9 ± 3.0% of CD45/Ter119/CD31negCxcl12-GFPhigh cells were tdTomato+, whereas 97.6 ± 1.1% of CD45/Ter119/CD31negCxcl12CE-tdTomato+ cells were Cxcl12-GFPhigh (*n* = 5, Fig. 1f, Supplementary Fig. 8a, b). Cxcl12CE-tdTomato+ cells ubiquitously expressed leptin receptor (LepR) (Fig. 1g, Supplementary Fig. 8a, b). Notably, Cxcl12CE-tdTomato+ cells were preferentially localized in the central marrow space (Fig. 1h). In fact, Cxcl12CE-tdTomato+ cells accounted for a majority of Cxcl12-GFPhigh and LepR+ stromal cells in the central marrow space (tdTomato+: 70.9 ± 15.5% and 70.3 ± 15.2% of total Cxcl12-GFPhigh and LepR+ cells, respectively). By contrast, Cxcl12CE-tdTomato+ cells accounted for only a small portion of Cxcl12-GFPhigh and LepR+ stromal cells in the

metaphyseal marrow space (tdTomato+: 3.6 ± 2.4% and 2.6 ± 1.7% of total Cxcl12-GFPhigh and LepR+ cells, respectively) or in the endosteal space (tdTomato+: 16.1 ± 6.2% and 12.6 ± 5.9% of total Cxcl12-GFPhigh and LepR+ cells, respectively, Fig. 1i), suggesting that Cxcl12CE-tdTomato+ cells represent a specific subset of LepR+CXCL12+ stromal cells that are distant from the bone surface. Moreover, *Cxcl12-creER* marked a subset of Cxcl12-GFPhigh cells that were characterized by the least mitotic activity and the most abundant expression of CXCL12 but not SCF (Fig. 1j–m, Supplementary Fig. 8a, b). Cxcl12CE-tdTomato+ cells were distinct from mature osteoblasts, as they did not express Col1a1(2.3 kb)-GFP (Fig. 1n, o, Supplementary Fig. 8a, b). Importantly, this *Cxcl12-creER* line had minimal promiscuity in the stromal cell compartment; although tdTomato+ cells were occasionally present in *Cxcl12-creER*; *R26R*tdTomato mice in the absence of tamoxifen (6/13 mice, 46%, Supplementary Fig. 1b, c), virtually all of them were in a CD45/Ter119/CD31+ fraction (90.9 ± 3.3%, Supplementary Figs. 1b, 8a). Therefore, *Cxcl12-creER* can mark a relatively quiescent subset of CXCL12+ perisinusoidal BMSCs in the central marrow space upon tamoxifen injection.

**Single-cell characterization of Cxcl12-creER+ BMSCs.** We further defined the identity of Cxcl12-creER+ stromal cells by a single cell RNA-seq analysis. To this end, we interrogated the profile of fluorescently sorted single cells gated on a GFPhigh fraction (Supplementary Fig. 8c, d) isolated from *Cxcl12*GFP/+; *Cxcl12-creER*; *R26R*tdTomato bone marrow at P28 after a tamoxifen pulse at P21 (Mouse #1: 2749 cells, Mouse #2: 4577 cells, total 7326 cells). Initial analysis revealed that a substantial fraction of cells (Mouse #1: 721/2749 cells, 26.2%, Mouse #2: 747/4577 cells, 16.3%) belonged to the clusters without detectable *eGFP* expression; these *eGFP*neg clusters included myeloid cells, lymphocytes, and erythroid cells (Supplementary Fig. 2), highlighting an issue on hematopoietic cell contamination commonly observed in recently published bone marrow stromal datasets[11–13]. *eGFP* was exclusively expressed by cells that abundantly expressed *Cxcl12* (Supplementary Fig. 2). Cxcl12-GFP+ cells were heterogeneous and clustered into nine groups, including three clusters of stromal (Clusters 0–2), two clusters endothelial (Clusters 4 and 8), one cluster of periosteal (Cluster 3) cells (Fig. 2a, Supplementary Fig. 2). Other small clusters included cells in cell cycle (Cluster 6) and enriched for mitochondrial (Cluster 5) and ribosomal (Cluster 7) genes. The stromal clusters were composed of a reticular cell group expressing pre-adipocyte markers such as *Adipoq* and *Lepr* (Cluster 0), and a group expressing pre-osteoblast markers such as *Alpl* and *Postn* (Cluster 1) (Fig. 2a, Supplementary Fig. 2). Cells in Cluster 0 were relatively enriched for secreted factors such as *Cxcl12*, *Kitl*, *Fgf7*, and *Il7*, whereas those in Cluster 1 were relatively enriched for an osteogenic factor *Clec11a*[14] (Fig. 2b). Importantly, tdTomato was identified as a cell-type specific marker for Cluster 0 (Fig. 2a, b). Thus, a majority of Cxcl12CE-tdTomato+ cells belongs to a reticular cluster in which cells abundantly express adipocyte-related genes and a variety of cytokines.

We next examined ex vivo properties of Cxcl12-creER+ stromal cells by a colony-forming unit fibroblast (CFU-F) assay (Supplementary Fig. 3a). To achieve this, we plated whole bone marrow cells isolated from *Cxcl12-creER*; *R26R*tdTomato mice at P28 (tamoxifen pulse at P21) at a clonal density. *Cxcl12-creER* marked only a small fraction of CFU-Fs (3.7 ± 0.8%, compared to 99.7 ± 0.6% of total CFU-Fs by ubiquitous *Ubc-creER*, Fig. 2c, d); this is in sharp contrast with *LepR-cre* and *Ebf3-creER* that can mark essentially all CFU-Fs[7,9]. Therefore, *Cxcl12-creER* can

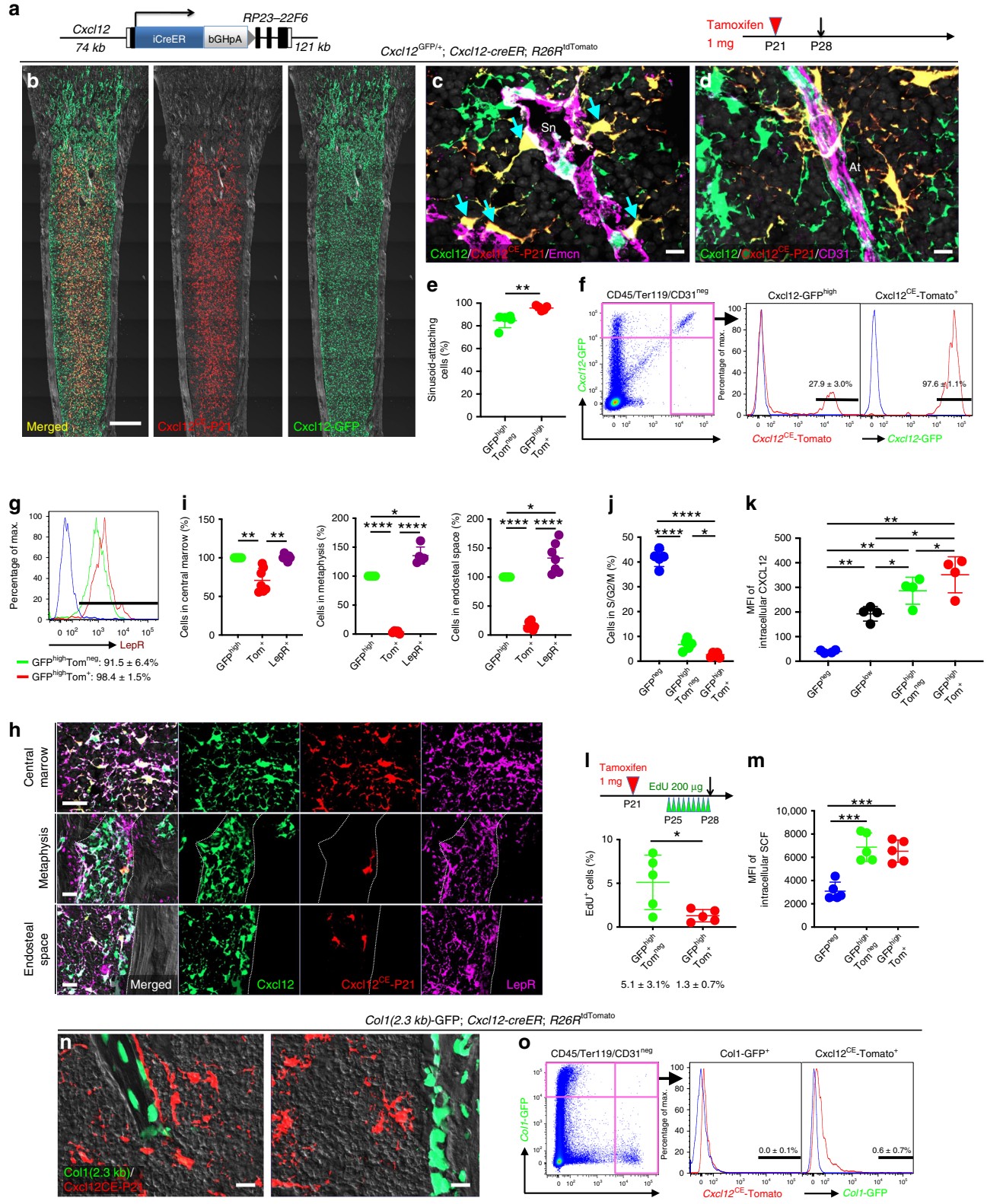

specifically mark a subset of CXCL12[+] BMSCs with little colony-forming activities upon tamoxifen injection.

Subsequently, we tested in vitro passageability of individual tdTomato[+] clones (Supplementary Fig. 3a). Cxcl12[CE]-tdTomato[+] clones could survive for significantly higher passages than Ubc[CE]-tdTomato[+] clones did; while 30.8% (8/26) of Cxcl12[CE]-tdTomato[+] clones could be passaged over

4 generations, only 8.3% (4/48) of Ubc[CE]-tdTomato[+] clones could be passaged over the same generation (Fig. 2e, Supplementary Fig. 3b). These Cxcl12[CE]-tdTomato[+] clones exhibited in vitro trilineage differentiation potential (i.e., adipocytes, osteoblasts, and chondrocytes, 12/12 clones, 100%, Fig. 3f), and differentiated into osteoblast-like cells depositing mineralized matrix upon transplantation into immunodeficient mice

**Fig. 1 Cxcl12-creER marks a quiescent subset of CXCL12+LepR+ BMSCs. a** Structure of *Cxcl12-iCreER-bGHpA* bacterial artificial chromosome (BAC) transgene. **b-o** Short-chase analysis of *Cxcl12-creER+* cells. **b** *Cxcl12*GFP/+; *Cxcl12-creER*; *R26R*tdTomato (pulsed at P21) distal femurs with growth plates on top. Gray: DIC. Scale bar: 500 μm. $n = 3$ mice. **c, d** Immunostaining for endomucin (Emcn, **c**) and CD31 (**d**). Sn: sinusoids, At: arterioles. Arrows: sinusoid-attaching GFPhightdTomato+ cells. Gray: DAPI. Scale bar: 10 μm. $n = 5$ mice. **e** Percentage of cells attaching Emcn+ sinusoids among GFPhightdTomatoneg (green) or GFPhightdTomato+ (red) cells. $n = 5$ mice. **f** Flow cytometry analysis of CD45/Ter119/CD31neg bone marrow cells. Blue lines: control cells. $n = 5$ mice. **g** Leptin receptor (LepR) expression based on flow cytometry analysis. Blue line: isotype control. $n = 3$ mice. **h** LepR staining. Gray: DAPI (Center), DIC (metaphysis and endosteal zone). Scale bar: 20 μm. **i** Quantification based on histology. Percentage of BMSCs expressing Cxcl12-GFP (green dots), Cxcl12CE-tdTomato (red dots) and LepR (purple dots). $n = 7$ fields from four mice (central marrow and endosteal surface), $n = 5$ fields from 4 mice (metaphysis). **j** Percentage of CD45/Ter119/CD31neg subpopulations in S/G2/M phase. $n = 5$ mice. **k** Intracellular CXCL12 protein levels of CD45/Ter119/CD31neg subpopulations. MFI: median fluorescence intensity. $n = 4$ mice. **l** EdU serial pulse assay of *Cxcl12-creER+* cells. *Cxcl12-creER*; *R26R*tdTomato mice were pulsed with tamoxifen at P21, then with EdU every 8 h for 3 days prior to analysis. Percentage of EdU+ cells among GFPhightdTomatoneg (green) or GFPhightdTomato+ (red) cells. $n = 5$ mice. **m** Intracellular SCF protein levels of CD45/Ter119/CD31neg subpopulations. MFI median fluorescence intensity. $n = 5$ mice. **n, o** *Col1*(2.3 kb)-GFP; *Cxcl12-creER*; *R26R*tdTomato. **n** Left panel: trabecular bone. Right panel: endosteal space. Gray: DIC. Scale bar: 20 μm. $n = 4$ mice. **o** Flow cytometry analysis of CD45/Ter119/CD31neg cells harvested from *Col1*(2.3 kb)-GFP; *Cxcl12-creER*; *R26R*tdTomato at P28 (pulsed at P21) bone marrow. Blue lines: control cells. $n = 4$ mice. ****$p < 0.0001$, ***$p < 0.001$, **$p < 0.01$, *$p < 0.05$, two-tailed, Mann–Whitney's U test (**e, l**). Two-tailed, one-way ANOVA followed by Tukey's post hoc test (**i-k, m**). All data are presented as mean ± s.d. Source data are provided as a Source Data file.

(Supplementary Fig. 3c). Thus, these small numbers of CFU-Fs marked by *Cxcl12-creER* possess robust in vitro self-renewability and a propensity to become osteoblasts in a non-native environment.

**Cell-fate analysis reveals dormancy of Cxcl12-creER+ BMSCs.** We subsequently defined long-term cell fates of quiescent Cxcl12-creER+ BMSCs using *Cxcl12-creER*; *R26R*tdTomato mice carrying either *Col1*(2.3 kb)-GFP or *Cxcl12*GFP/+ reporters. These mice were pulsed at P3, P21, and 8 weeks of age (hereafter, Cxcl12CE-P3, Cxcl12CE-P21, and Cxcl12CE-8 W cells, respectively). Cxcl12CE-P3 cells remained spatially restricted only in a far proximal portion of the femoral marrow space at 1 year of age (1 y) (Fig. 3a). Similarly, few Cxcl12CE-P21 cells migrated distally to contribute to newly formed marrow space beneath the growth plate toward 1 y (Fig. 3b, Supplementary Figs. 4a–g and 8a, b). By contrast, Cxcl12CE-8 W cells were distributed throughout the marrow space at 12 W and 1 y (Fig. 3c, Supplementary Fig. 4h), demonstrating that a tamoxifen pulse at the adult stage could label cells under the growth plate that were not labeled when tamoxifen was given at earlier stages (Fig. 3d). Therefore, these Cxcl12-creER+ BMSCs are dormant and remain in the original domain of the marrow space in growing bones, without producing new stromal cells in the newly formed domain of the marrow space in the metaphyseal region resulting from long-bone growth. These data also indicate that quiescent Cxcl12-creER+ BMSCs might be produced by other mesenchymal precursor cells during bone growth, such as type II collagen (Col2a1)+ cells residing in proximity to the growth plate or parathyroid hormone-related protein (PTHrP)+ resting chondrocytes as demonstrated previously[15,16].

Cxcl12-creER+ BMSCs marked by *Cxcl12-creER* remained in the marrow space predominantly as Cxcl12-GFPhigh reticular cells (Fig. 3e). Quantification of Cxcl12CE-8 W cells revealed that these cells did not change in number within the marrow space over time through 1 y 6 M (Fig. 3f). Importantly, *Cxcl12-creER*-marked stromal cells, regardless of the time they were marked ((i.e., Cxcl12CE-P3, Cxcl12CE-P21, Cxcl12CE-8 W, and Cxcl12CE-5 M cells), made virtually no contribution to cortical osteoblasts during the prolonged chase until 1 y 6 M (Fig. 3g–l, Supplementary Fig. 8a, b). Interestingly, these cells contributed to a fraction of trabecular osteoblasts/cytes (Fig. 3j–l). The fraction of Cxcl12CE-8 W cells among Col1(2.3 kb)-GFP+ trabecular osteoblasts/cytes increased progressively during the first 6 months of chase and reached the plateau thereafter (Fig. 3l red line). By contrast, essentially no Cxcl12CE-8 W cells contributed to cortical osteoblasts/cytes during the entire duration of chase (Fig. 3l, blue line). Therefore, quiescent Cxcl12-creER+ perisinusoidal BMSCs predominantly stay within the marrow space as CXCL12+ stromal cells and do not migrate and differentiate into cortical bone osteoblasts in homeostasis.

**Cortical bone regenerative responses of Cxcl12-creER+ BMSCs.** Subsequently, we asked whether quiescent Cxcl12-creER+ BMSCs can participate in skeletal regeneration. The *Cxcl12-creER* line described here gives us the opportunity to experimentally interrogate their potential to participate in cortical bone regeneration, as quiescent Cxcl12-creER+ BMSCs do not normally differentiate into cortical bone osteoblasts. For this purpose, we utilized a drill-hole cortical injury model[17] (Fig. 4a) that disrupts the femoral cortical bone and the endosteal surface to induce cortical bone regeneration through intramembranous ossification. We performed the surgery at 1 week after *Cxcl12-creER*; *R26R*tdTomato mice carrying either *Col1*(2.3 kb)-GFP or *Cxcl12*GFP/+ reporter were pulsed at 6–10 weeks of age (Fig. 4b). After 2 days of injury, a substantial portion of Cxcl12CE-tdTomato+ cells adjacent to the injury site proliferated (EdU+ cells: $59.1 \pm 10.8\%$ of tdTomato+ cells, compared to $0.3 \pm 0.7\%$ in the intact site, $n = 7$), and invaded into the defect while maintaining Cxcl12-GFP expression (Fig. 4c–e). Cxcl12CE-tdTomato+ cells continued to increase at the injury site, and occupied the inter-trabecular space of newly formed woven bones (hard callus) in the cortical bone after seven days of injury (Day 7, Fig. 4f); particularly those within the hard callus started to express Col1 (2.3 kb)-GFP at this stage (Fig. 4g). After two weeks of injury, a majority of Cxcl12CE-tdTomato+ cells differentiated into Col1 (2.3 kb)-GFP+ osteoblasts within the healing cortical bone (Day 14, Fig. 4h, i). After 8 weeks of injury, Cxcl12CE-tdTomato+ cells robustly contributed to osteocytes in the regenerated portion of the cortical bone (Day 56, Fig. 4j, k), which was not observed in the intact portion of the cortical bone (Cxcl12CE-tdTomato+ osteocytes: $0.1 \pm 0.2\%$ in Intact, $39.7 \pm 1.9\%$ in Injured, Fig. 4l, m). Therefore, quiescent Cxcl12-creER+ BMSCs in the marrow space can be recruited to a cortical defect in response to injury and differentiate into osteoblasts/cytes during cortical bone repair, unlike in homeostasis.

Multiple types of mesenchymal cells can participate in bone repair by becoming osteoblasts[18]. To address whether other osteoblast precursor cells can participate in cortical bone regeneration, we further took advantage of an *osterix (Osx)-creER* line[19]. We performed the surgery at 1 week after *osteocalcin (Ocn)-GFP*; *Osx-creER*; *R26R*tdTomato mice were pulsed at 6–10 weeks of age, and analyzed these mice after 8 weeks of injury. OsxCE-tdTomato+ cells moderately contributed to

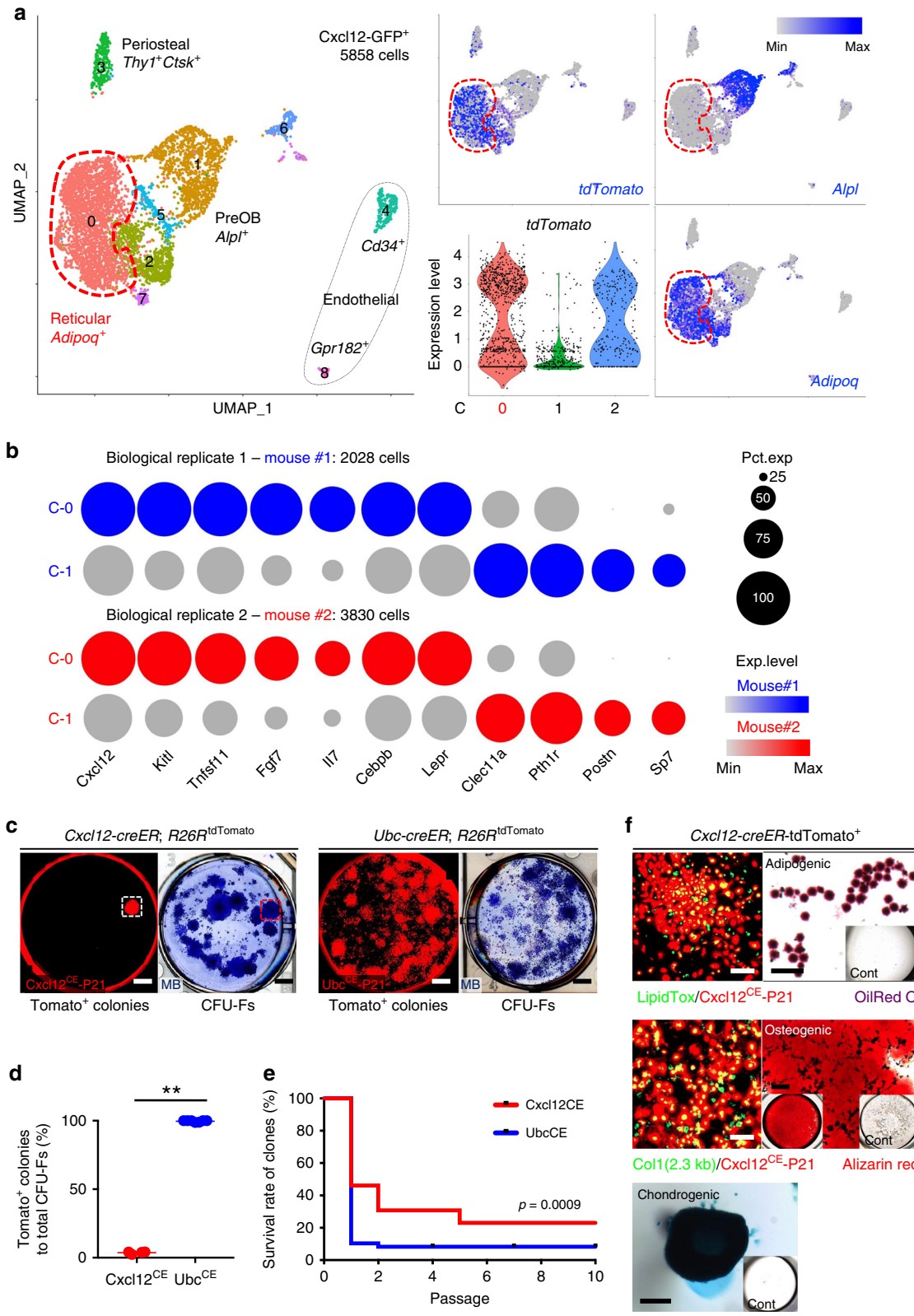

osteocytes in the regenerated portion of the cortical bone (Osx[CE]-tdTomato[+] osteocytes: 12.3 ± 4.9%, Fig. 4m, n). These data suggest that Osx-creER[+] cells might be one of other progenitor populations that contribute to approximately 60% of osteocytes in the regenerated portion that are not derived from Cxcl12-creER[+] BMSCs.

As expected, *Cxcl12-creER*-marked stromal cells readily became marrow adipocytes. Cxcl12[CE]-8 W and Cxcl12[CE]-5 M cells spontaneously differentiated into Perillipin[+] marrow

**Fig. 2 Single-cell characterization of Cxcl12-creER[+] BMSCs. a** Single-cell RNA-seq analysis, UMAP-based visualization of major classes of Cxcl12-GFP[+] cells (Clusters 0–8). Red dotted contour: reticular cells (Cluster 0). Center panels: *tdTomato* expression, feature plot (top), violin plot (Clusters 0–2) (bottom). Right panels: feature plots. Blue: high expression. $n = 5858$ cells merged from two biological replicates (Mouse #1: 2028 cells, Mouse #2: 3830 cells). **b** Split-dot-based visualization of representative gene expression in Cluster 0 (C-0, reticular cells) and Cluster-1 (C-1, pre-osteoblasts). Upper: Mouse #1, Lower: Mouse #2. Circle size: percentage of cells expressing a given gene in a given cluster (0–100%), color density: expression level of a given gene. **c–f** CFU-F assay. **c** Cxcl12[CE]-tdTomato[+] (left) and Ubc[CE]-tdTomato[+] (right) bone marrow cells at P28 (pulsed at P21). Scale bar: 5 mm. **d** percentage of tdTomato[+] colonies among total CFU-Fs. $n = 4$ (Cxcl12[CE]), $n = 7$ (Ubc[CE]) mice. **$p < 0.01$, two-tailed, Mann–Whitney's $U$ test. Data are presented as mean ± s.d. **e** Survival curve of individual tdTomato[+] clones over serial passages. $n = 26$ (Cxcl12[CE]), $n = 48$ (Ubc[CE]) clones. Gehan–Breslow–Wilcoxon test. **f** In vitro trilineage differentiation assay of Cxcl12[CE]-tdTomato[+] clones (Passage 2–7). Upper panels: Adipogenic conditions. LipidTOX staining (left). Green: LipidTOX-Alexa488, red: tdTomato. Oil red O staining (Right). Center panels: Osteogenic conditions. Green: Col1(2.3 kb)-GFP, red: tdTomato. Alizarin Red staining (right). Lower panel: Chondrogenic conditions, Alcian Blue staining. Insets: differentiation medium negative controls. Scale bar: 200 μm. $n = 12$ clones. Source data are provided as a Source Data file.

adipocytes after the long chase, at 3 M and 1 y 6 M, respectively (Supplementary Fig. 5a). In addition, *Cxcl12-creER[+]* reticular cells became marrow adipocytes in the presence of high-fat diet containing PPARγ agonist rosiglitazone (Rosi) (LipidTOX[+]tdTomato[+] cell: Rosi, 39.3 ± 14.6 vs. Cont, 2.5 ± 1.0 cells per section, Supplementary Fig. 5b–e). A committed adipogenic subset of LepR[+]CXCL12[+] cells in the marrow space is involved in bone fracture healing[20]. We, therefore, performed a tibial complete fracture experiment, a defect of which is repaired primarily through endochondral ossification. We performed the surgery at 1 week after *Col1(2.3 kb)-GFP; Cxcl12-creER; R26R[tdTomato]* mice were pulsed at 6–10 weeks of age (Supplementary Fig. 5f). After 2 weeks of fracture, Cxcl12[CE]-tdTomato[+] cells increased substantially within the fracture callus and became a number of Sox9[+] cells (Supplementary Fig. 5g), indicating that Cxcl12-creER[+] cells can differentiate into chondrocytes during fracture healing.

**Injury-induced identity conversion of Cxcl12-creER[+] BMSCs.** We subsequently set out to define the mechanism underlying the regenerative capacity of Cxcl12-creER[+] BMSCs. For this purpose, we investigated how injury induces change in cell populations and their transcriptomes of quiescent Cxcl12-creER[+] BMSCs by combined lineage-tracing and single cell RNA-seq analyses. As an injury model, we made use of a femoral bone marrow ablation model that mechanically removes a defined area of bone marrow using an endodontic instrument[21] (Fig. 5a, b). This procedure induces direct differentiation of marrow stromal cells within the marrow cavity without involving the periosteum. The right femur underwent the surgery, while the left femur was untreated and used as an internal control (Fig 5a). Marrow ablation induced differentiation of Cxcl12[CE]-tdTomato[+] cells into Col1a1(2.3 kb)-GFP[+] osteoblasts within the marrow space (Fig. 5c–j, Supplementary Fig. 8a, b).

We interrogated the single-cell profile of lineage-marked Cxcl12[CE]-tdTomato[+] cells from the ablated (ABL) and the contralateral control femur (CONT) individually after 7 days of marrow ablation, followed by integration of two datasets using Seurat/CCA[22] (Fig. 6a, Supplementary Fig. 8c, e). Initial analysis revealed that a substantial fraction of cells belonged to the clusters without detectable *tdTomato* expression; these *tdTomato[neg]* clusters included myeloid cells, lymphocytes and erythroid cells (Supplementary Fig. 6a). *tdTomato* was predominantly expressed by mesenchymal cells expressing *Cxcl12* or *Col1a1*, but also by a small number of endothelial cells. A graph-based clustering analysis of integrated mesenchymal Cxcl12[CE]-tdTomato[+] cells revealed 12 clusters, including three major groups of *Cxcl12[+]Adipoq[+]* reticular cells (Clusters 0–2) and small groups of *Alpl[+]* osteoblastic cells (Clusters 4 and 6); a terminally differentiated osteoblast marker *Ifitm5* was exclusively expressed by cells in Cluster 4 (Fig. 6b). Most notably, marrow ablation

appeared to induce a shift in cell populations toward pre-osteoblasts and osteoblasts; ABL cells were more prominent in these osteoblastic clusters (Fig. 6c, yellow arrowheads). Concomitantly, cells in both reticular and pre-osteoblastic clusters appeared to undergo upregulation of osteoblast-signature genes in response to injury (Fig. 6d). Analysis of the second biological replicate showed a pattern similar to the first replicate (Supplementary Fig. 6a, b). In addition, a colony-forming assay revealed that marrow ablation induced a significant increase in Cxcl12[CE]-tdTomato[+] CFU-Fs ex vivo (tdTomato[+] colony: CONT: 1.4 ± 1.1% vs. ABL: 10.5 ± 6.5% of total CFU-Fs, Fig. 6e, f), consistent with a shift in cell populations toward pre-osteoblasts in the ablated marrow identified by the single cell RNA-seq analysis. Therefore, Cxcl12-creER[+] BMSCs undergo a transcriptional change that drives them for an osteoblast-like state in response to injury, associated with an increase in colony-forming activities.

**Cxcl12-creER[+] BMSCs transform into skeletal stem-like cells.** Further, we set out to define the functional importance of Cxcl12-creER[+] BMSCs in cortical bone regeneration by inducible cell ablation experiments using a *Rosa26-iDTA* (inducible diphtheria toxin fragment A) allele. Flow cytometry analysis of *Cxcl12-creER; Rosa26[lsl-tdTomato/+]* (tdT) and *Cxcl12-creER; Rosa26[lsl-tdTomato/iDTA]* (DTA) littermates at 16 weeks of age after being pulsed at 4, 6, and 8 weeks of age revealed a 6.8-fold reduction of tdTomato[+] cells in DTA mice (tdTomato[+] cells: tdT, 0.045 ± 0.004% vs. DTA, 0.007 ± 0.006% of CD45/Ter119/CD31[neg] fraction, Fig. 7a, Supplementary Fig. 8a, $n = 3$ for Control, $n = 6$ for DTA mice), demonstrating the effectiveness of an iDTA-mediated cell ablation approach. Subsequently, we pulsed *Rosa26[lsl-tdTomato/iDTA]* (Control) and *Cxcl12-creER; Rosa26[lsl-tdTomato/iDTA]* (DTA) littermates with tamoxifen at 4 weeks of age, and performed the surgery at 1 week after the pulse. After 14 days of injury, bone volume and bone mineral density of the injured cortical area were significantly reduced in DTA mice (Fig. 7b, c). Therefore, quiescent Cxcl12-creER[+] BMSCs are a functionally important contributor to cortical bone regeneration.

Further, we aimed to define how quiescent Cxcl12-creER[+] BMSCs can be activated and embark on the differentiation trajectory into osteoblasts. For this purpose, we examined the single cell profile of lineage-marked Cxcl12[CE]-tdTomato[+] cells from the ablated femur after 14 days of marrow ablation (Fig. 7d, Supplementary Fig. 8c, e). Initial analysis revealed again that a substantial fraction of cells (7074/9469 cells) belonged to the clusters without detectable *tdTomato* expression. A graph-based clustering analysis of mesenchymal Cxcl12[CE]-tdTomato[+] cells revealed that pre-osteoblasts (preOBs) appeared as an intermediate state between *Cxcl12[+]Adipoq[+]* perisinusoidal reticular cells and *Bglap[+]* mature osteoblasts (Fig. 7e, Supplementary Fig. 7a–c). These intermediate-state cells expressed *Itgav*

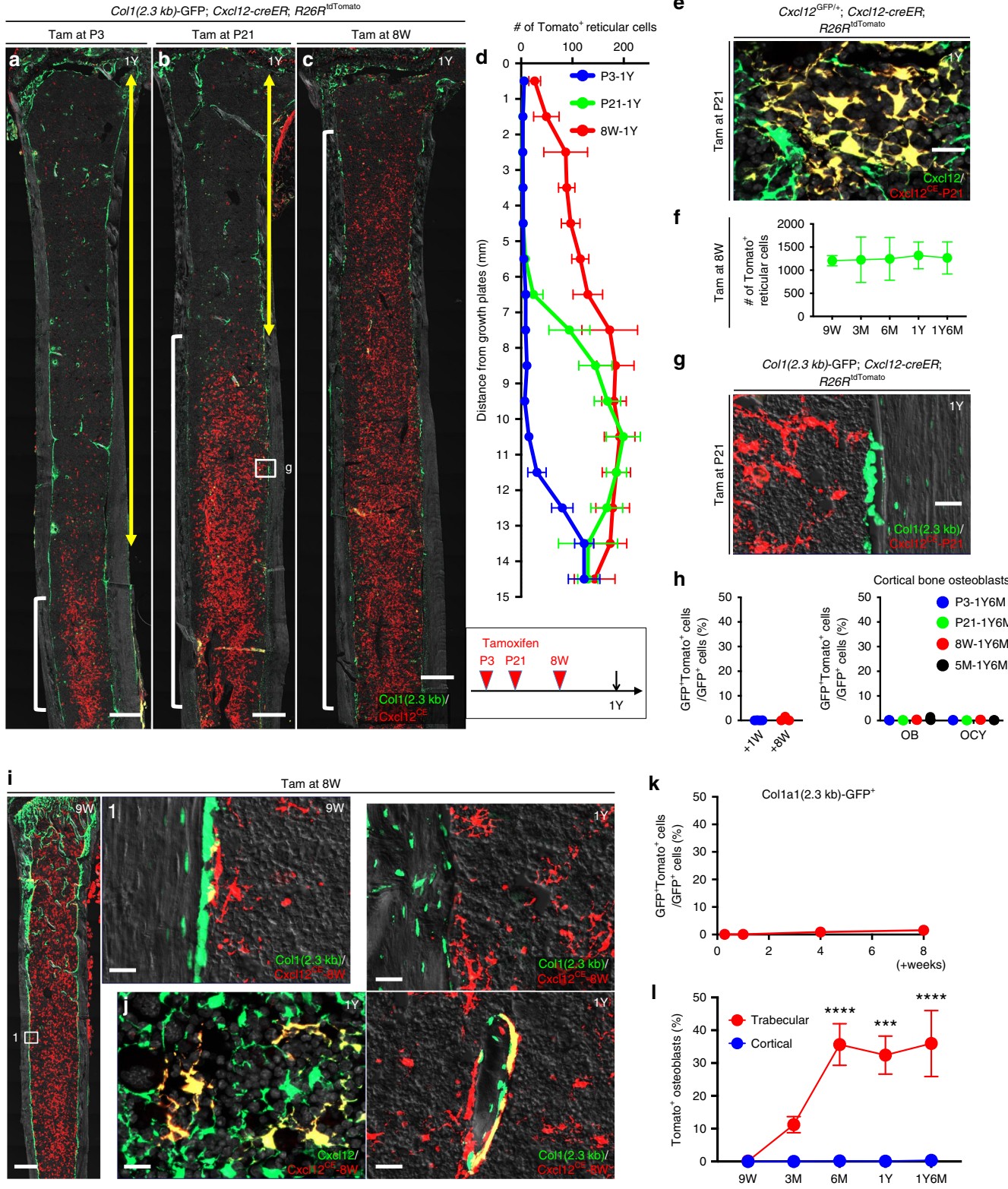

(encoding CD51) and *Cd200* (CD200), but not *Thy1* (CD90), *Eng* (CD105), *Grem1* (Grem1), or *Ly6a* (Sca1), indicating that these cells might overlap with previously reported mouse SSCs (mSSCs)[23], but not with osteo-chondro-reticular stem cells[24] (Fig. 7f). We further constructed an unsupervised single-cell trajectory using Monocle (Supplementary Fig. 7d). This analysis predicted a continuous lineage path without a branch, and empirically identified pseudotime-dependent genes that were

highly expressed in populations corresponding to perisinusoidal reticular cells, pre-osteoblasts and mature osteoblasts (Fig. 7g, h). Functional annotation of 970 pseudotime-dependent differentially expressed genes using DAVID revealed significant enrichment of Gene Ontology terms related to Wnt signaling (3 out of top 7 most significant terms, Wnt signaling pathway [GO:0016055], negative regulation of Wnt [GO:0030178] and canonical Wnt [GO:0090090] signaling pathway) (Fig. 7i). In fact,

**Fig. 3 Cell-fate analysis reveals dormancy of Cxcl12-creER+ BMSCs. a–l** Long-chase analysis of *Cxcl12-creER+* cells. **a–c** *Col1*(2.3 kb)-GFP; *Cxcl12-creER*; *R26R*tdTomato distal femurs with growth plates on top, analyzed at 1Y. Pulsed at P3 (**a**), P21 (**b**), and 8 W (**c**). Yellow double arrows: tdTomatoneg area beneath growth plates, white brackets: tdTomato+ domain in marrow space. Scale bar: 500 μm. Gray: DIC. **d** Quantification of tdTomato+ reticular cells in marrow space at 1Y, based on distance from growth plate. Pulsed at P3 (blue line), P21 (green line) or 8 W (red line). $n = 6$ (P3, P21-pulsed), $n = 5$ (8W-pulsed) mice. **e** *Cxcl12*GFP/+; *Cxcl12-creER*; *R26R*tdTomato bone marrow at 1 y (pulsed at P21), central marrow space. Gray: DAPI. Scale bar: 20 μm. $n = 3$ mice. **f** Quantification of tdTomato+ reticular cells in marrow space during extended chase, pulsed at 8 W. $n = 4$ (9 W), $n = 5$ (3 M), $n = 3$ (6 M, 1 y, and 1 y 6 M) mice. **g** *Col1*(2.3 kb)-GFP; *Cxcl12-creER*; *R26R*tdTomato bone marrow at 1 y (pulsed at P21), endosteal space. Scale bar: 20 μm. $n = 4$ mice. **h** Percentage of Col1(2.3 kb)-GFP+tdTomato+ cells per total Col1(2.3 kb)-GFP+ cortical bone osteoblasts. (Left): flow cytometry. $n = 4$ (+1 W), $n = 3$ (+8 W) mice. (Right): labeled at P3, P21, 8 W, and 5 M, analyzed at 1 y 6 M. OB osteoblasts, OCY osteocytes. $n = 3$ per group. **i** *Col1*(2.3 kb)-GFP; *Cxcl12-creER*; *R26R*tdTomato distal femurs at 9 W (pulsed at 8 W). 1 (endosteal area): boxed area on left panel. Gray: DIC. Scale bar: 500 μm (left), 20 μm (right). $n = 3$ mice. **j** *Cxcl12-creER*; *R26R*tdTomato distal femur bone marrow at 1Y (pulsed at 8 W). Lower left panel: central marrow space with *Cxcl12*GFP/+, right panels: endosteal space and trabecular bone with *Col1*(2.3 kb)-GFP. Gray: DIC. Scale bar: 20 μm. $n = 3$ mice. **k, l** Percentage of Col1(2.3 kb)-GFP+tdTomato+ cells per total Col1(2.3 kb)-GFP+ cells by using flow cytometry (**k**). $n = 3$ (+2 days), $n = 4$ (+1 W), $n = 7$ (+4 W) and $n = 6$ (+8 W) mice. **l** Labeled at 8 W, analyzed at 9 W, 3 M, 6 M, 1 y, and 1 y 6 M. $n = 3$ (9 W, 3 M, 1 y, and 1 y 6 M), $n = 4$ (6 M) mice. ****$p < 0.0001$, ***$p < 0.001$, two tailed, one-way ANOVA followed by Tukey's post hoc test (**f**, **h**, **k**, **l**). Two-tailed, Mann–Whitney's *U* test (**h**). All data are presented as mean ± s.d. Source data are provided as a Source Data file.

---

genes encoding soluble Wnt antagonists including *Sfrp1*, *Sfrp2* and *Sfrp4* were enriched in perisinusoidal reticular cells, whereas *Ctnnb1*, a key component of Wnt/β-catenin signaling, was enriched in pre-osteoblasts. Therefore, quiescent Cxcl12-creER+ BMSCs appear to undergo activation of canonical Wnt signaling during their transition into pre-osteoblasts.

**Wnt-mediated transformation orchestrates bone regeneration.** To test the hypothesis that canonical Wnt signaling plays an important role in this transition, we conditionally deleted β-catenin (*Ctnnb*) in Cxcl12-creER+ BMSCs during cortical bone regeneration, using *Cxcl12-creER*; *Ctnnb*+/+; *R26R*tdTomato (Control), *Cxcl12-creER*; *Ctnnb*fl/+; *R26R*tdTomato (βCat cHet) and *Cxcl12-creER*; *Ctnnb*fl/fl; *R26R*tdTomato (βCat cKO) littermates. After 14 days of cortical bone injury, bone volume and bone mineral density of the injured cortical area were significantly reduced in βCat cKO mice compared with Control or βCat cHet mice, associated with reduction in alkaline phosphatase (ALP)+ cells (Fig. 8a, b). The reduction of bone volume in βCat cKO mice was comparable to that of DTA mice, indicating that canonical Wnt signaling plays essential functional roles in Cxcl12-creER+ BMSCs' differentiation trajectory toward osteoblasts.

Transcription factors Sox9 and Runx2 cooperatively regulate commitment to the osteoblast lineage in a manner regulated by canonical Wnt signaling during skeletal development[25,26]. However, our analysis showed that *Sox9* expression was unaltered, while *Runx2* expression was not consistently altered in response to injury (Fig. 6d). To test whether Sox9 and Runx2 play roles in cell identity conversion, we conditionally deleted Sox9 or Runx2 in quiescent Cxcl12-creER+ BMSCs during cortical bone regeneration, using *Cxcl12-creER* and floxed alleles for *Sox9* and *Runx2*. After 14 days of cortical bone injury, bone volume and bone mineral density of the injured cortical area was not altered in Sox9 cKO mice or in Runx2 cKO mice (Fig. 8c), indicating that their transition into pre-osteoblasts is not regulated by Sox9 or Runx2.

Further, to address whether high levels of β-catenin can inhibit osteoblast differentiation of Cxcl12-creER+ BMSCs during cortical bone regeneration, we conditionally induced haploinsufficiency of *adenomatous polyposis coli* (*Apc*), which is a critical component of the β-catenin degradation complex, during cortical bone regeneration using *Cxcl12-creER*; *Apc*+/+; *R26R*tdTomato (Control) and *Cxcl12-creER*; *Apc*fl/+; *R26R*tdTomato (Apc cHet) littermates. After 7 days of cortical bone injury, there was no significant change in bone volume and bone mineral density of the injured cortical area in Apc cHet mice (Fig. 8d). Therefore, a high level of β-catenin in Cxcl12-creER+ BMSCs due to APC

haploinsufficiency does not inhibit their osteoblast differentiation during cortical bone regeneration. A limitation of this approach is that it provides a different level of β-catenin stabilization than removing the phosphorylation sites encoded by exon 3.

Canonical Wnt signaling plays disparate roles in different phases of bone fracture repair[27]. To determine whether Cxcl12-creER+ cells are more or less sensitive to β-catenin than other fracture repair sources, we conditionally deleted *Ctnnb* in osteoblast precursor cells during cortical regeneration, using *Ctnnb*fl/fl; *R26R*tdTomato (Control) and *Dlx5-creER*; *Ctnnb*fl/fl; *R26R*tdTomato (Dlx5-βCat cKO) littermates. After 14 days of cortical bone injury, bone volume and bone mineral density of the injured cortical area were significantly reduced Dlx5-βCat cKO mice compared with Control mice (Fig. 8e, f). Therefore, Cxcl12-creER+ BMSCs are equally as sensitive to β-catenin as Dlx5-creER+ osteoblast precursor cells.

Taken together, our findings provide functional evidence that quiescent Cxcl12-creER+ BMSCs orchestrates cortical bone regeneration in a unique manner mediated by canonical Wnt signaling independent of Sox9 or Runx2 function.

## Discussion

Here, we identified that quiescent Cxcl12-creER+ BMSCs can convert their identity into a precursor cell state similar to SSCs during injury responses in a manner mediated by canonical Wnt signaling, and substantially contribute to skeletal regeneration (see Fig. 9a–c). Importantly, these Cxcl12-creER+ BMSCs show little colony-forming activities and do not contribute to cortical bone osteoblasts in homeostasis or new CXCL12+ cells in growing marrow space, therefore representing a specific quiescent subpopulation of LepR+CXCL12+ BMSCs marked by *LepR-cre* or *Ebf3-creER*[7,9]. LepR+CXCL12+ cells that contribute to cortical bone osteoblasts in physiological conditions are within the majority of perisinusoidal cells that are not marked by *Cxcl12-creER*. Injury-induced cues drive these Cxcl12-creER+ BMSCs into a reticular-osteoblast hybrid state associated with global upregulation of osteoblast-signature genes. Our findings support the theory that canonical Wnt-mediated cellular plasticity underpins the remarkable regenerative capacity of CXCL12+ BMSCs. Canonical Wnt signaling is widely known as the central regulator of osteogenesis, particularly during early skeletal development[25,26]. In adult bones, canonical Wnt signaling prevents a cell fate shift of preosteoblasts to adipocytes[28] and inhibits excessive osteoclastogenesis through regulating mature osteoblasts[29]. In addition, liposomal delivery of Wnt proteins can promote bone regeneration[30]. Our findings highlight a mechanism that canonical Wnt signaling facilitates osteogenesis by

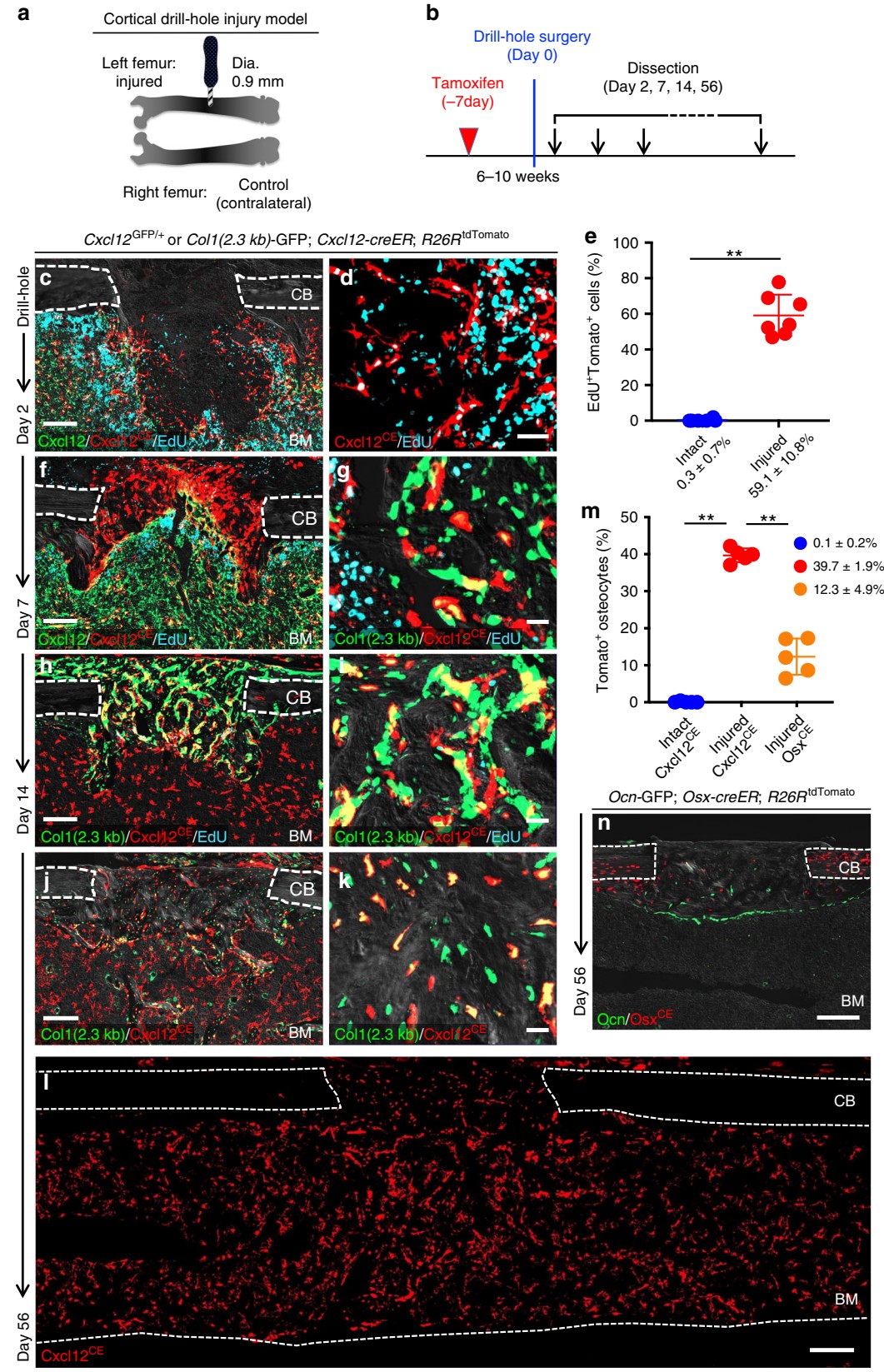

transforming CXCL12$^+$ BMSCs into osteoblast precursor cells particularly under regenerative conditions. CXCL12$^+$ BMSCs are equally as sensitive to deficiency in canonical Wnt signaling as other osteoblast precursor cells. This Wnt-mediated mechanism

appears to be unique because, unlike in skeletal development, it does not mediate Sox9 or Runx2 function.

The remarkable regenerative potential of bones is generally attributed to rare SSCs. Our findings raise an intriguing possibility that the previously defined SSCs[23] can be generated through

**Fig. 4 Cortical bone regenerative responses of Cxcl12-creER⁺ BMSCs.** (**a–n**) Fates of *Cxcl12-creER⁺* cells or *Osx-creER⁺* cells during cortical bone regeneration after drill-hole injury. **a** Experimental schemes for drill-hole cortical injury model. **b** Timeline for tamoxifen injection, drill-hole injury and analysis. *Cxcl12*GFP/⁺; *Cxcl12-creER*; *R26R*tdTomato (**c**, **f**) or *Col1*(2.3 kb)-GFP; *Cxcl12-creER*; *R26R*tdTomato (**g–k**) femur diaphyseal cortical bones (pulsed at 7 days before surgery) with periosteum on top. CB cortical bone, BM bone marrow. Gray: DIC. n = 5 mice per each group. **c**, **d** Injured area at 2 days after injury, femurs with EdU administration shortly before analysis. Scale bar: 200 μm (**c**), 50 μm (**d**). **e** Quantification of EdU⁺tdTomato⁺ cells. n = 6 (Intact), n = 7 (Injured). EdU was administered twice (6 and 3 h) prior to analysis. **p < 0.01, two-tailed Mann–Whitney's U test. **f**, **g** Injured area at 7 days after injury, femurs with EdU administration shortly before analysis. Scale bar: 200 μm (**f**), 20 μm (**g**). **h**, **i** Injured area at 14 days after injury, femurs with EdU administration shortly before analysis. Scale bar: 200 μm (**h**), 20 μm (**i**). **j-l** Injured cortical bone at 56 days after injury. Scale bar: 200 μm (**h**, **l**), 20 μm (**i**). **m** Number of tdTomato⁺ cells within intact and injured are of *Cxcl12-creER*; *R26R*tdTomato cortical bone (blue and red dots, respectively) or injured area of *Osx-creER*; *R26R*tdTomato cortical bone (orange dots). n = 5 mice per group. **p < 0.01, two-tailed, Mann–Whitney's U test. **n** Injured cortical bone at 56 days after injury, *Ocn-GFP*; *Osx-creER*; *R26R*tdTomato femurs. All data are presented as mean ± s.d. Source data are provided as a Source Data file.

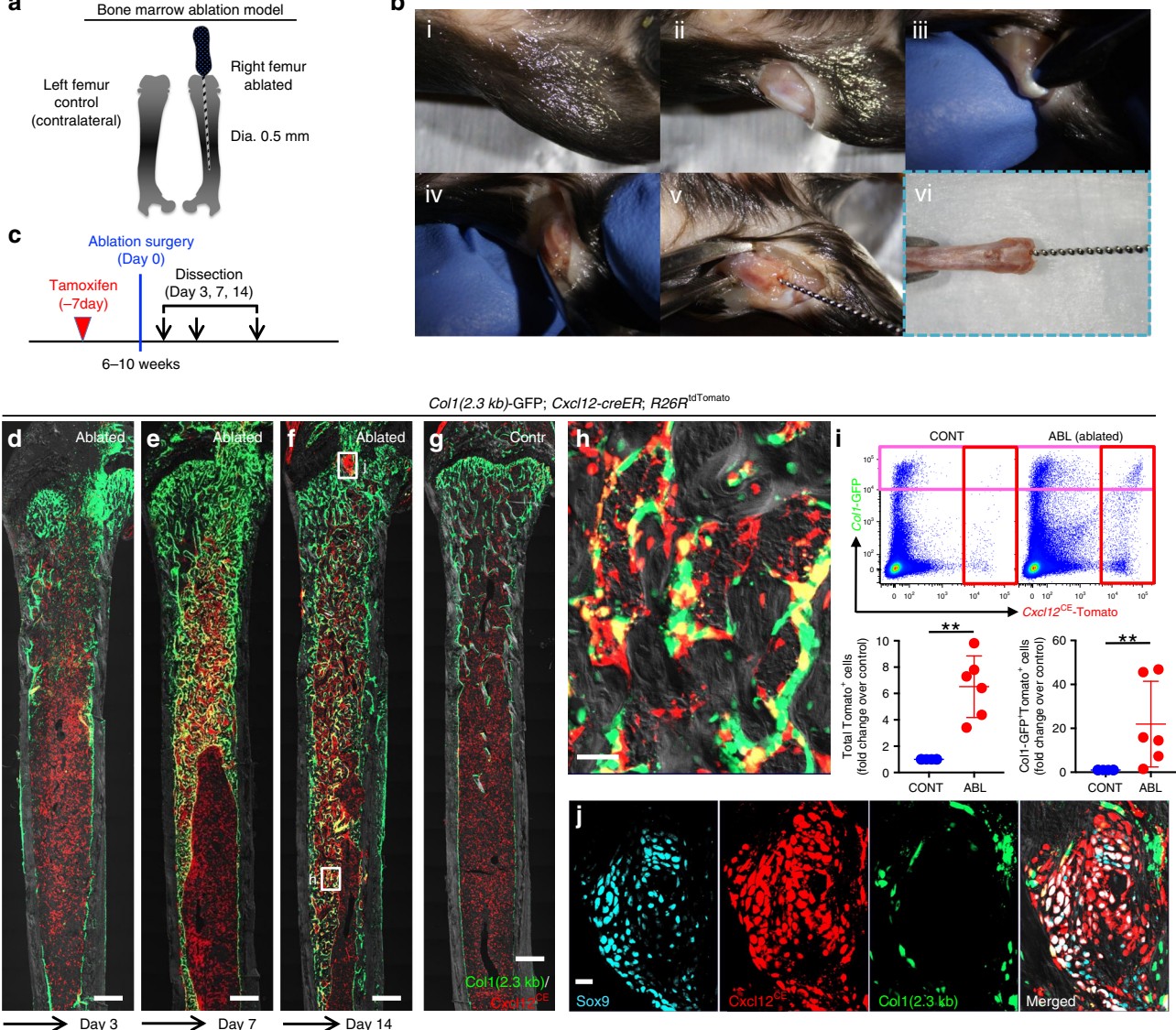

**Fig. 5 Cxcl12-creER⁺ BMSCs respond to bone marrow ablation-induced osteogenesis. a–c** Experimental schemes and procedures of bone marrow ablation model. **a** Right femurs were operated, while left femurs were unoperated and used as internal control. (**b**, i–vi): Step-by-step demonstration of surgical procedures. **c** Timeline for tamoxifen injection, ablation surgery and analysis. **d–j** *Col1*(2.3 kb)-GFP; *Cxcl12-creER*; *R26R*tdTomato distal femur bone marrow (pulsed at 7 days before surgery) with growth plates on top. Three days (**d**), 7 days (**e**), and 14 days after surgery (**f**, **h**). **h** Ablated area, boxed area in (**f**). Scale bar: 50 μm. **g** contralateral control side. Gray: DIC. Scale bar: 500 μm. n = 4 (Day 3), n = 3 (Day 7), n = 5 (Day 14), n = 3 (Conrol) mice. **i** Flow cytometry analysis of CD45/Ter119/CD31neg bone marrow cells. Lower panels: fold change of total tdTomato⁺ cells (left) or Col1-GFP⁺tdTomato⁺cells (right) in ablated side over contralateral control side. Scale bar: 20 μm. n = 6 mice. **p < 0.01, two-tailed, Mann–Whitney's U test. Data are presented as mean ± s.d. **j** Regenerated cartilaginous tissue adjacent to growth plate. Gray: DIC. Source data are provided as a Source Data file.

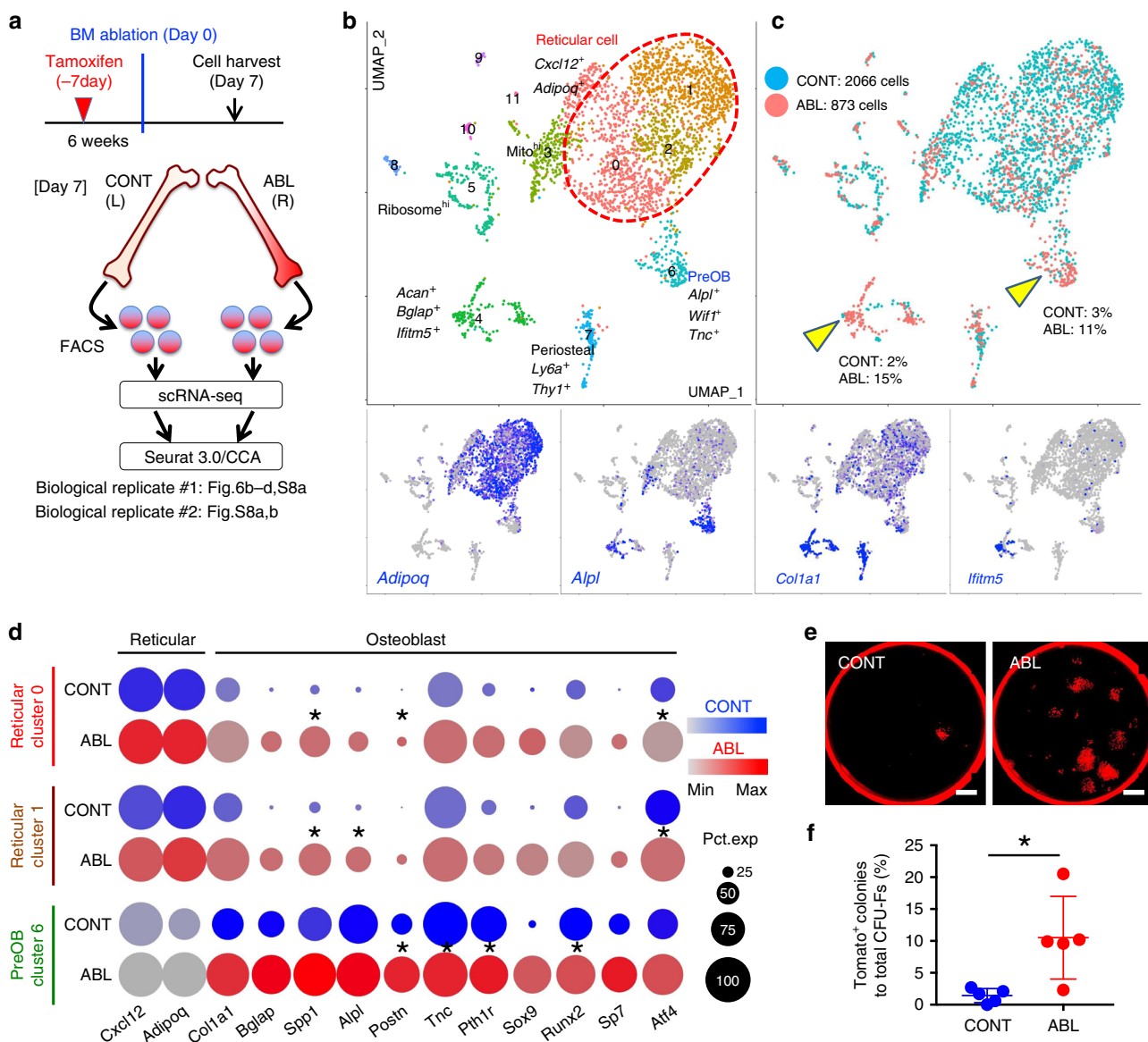

**Fig. 6 Injury-induced identity conversion of Cxcl12-creER⁺ BMSCs. a–d** Single-cell RNA-seq analyses of lineage-traced *Cxcl12-creER⁺* cells. **a** Cxcl12$^{CE}$-tdTomato⁺ cells were isolated from contralateral (CONT) and ablated (ABL) femurs separately at 7 days after marrow ablation. Two datasets were integrated by Seurat/CCA. **b**, **c** UMAP-based visualization of major classes of mesenchymal Cxcl12$^{CE}$-tdTomato⁺ cells (Cluster 0–11). Left panel, red dotted contour: reticular (Clusters 0–2). Lower panels: feature plots. Blue: high expression. Right panel: cells colored by conditions (CONT, ABL). Biological replicate 1, $n = 2939$ cells (CONT: 2066 cells, ABL: 873 cells), pooled from $n = 7$ mice. **d** Split-dot-based visualization of representative gene expression. Clusters 0 and 1 (reticular) and Cluster 6 (pre-osteoblast) shown for reticular-signature genes (Cxcl12, *Adipoq*) and osteoblast-signature genes (*Col1a1*, *Bglap*, *Spp1*, *Alpl*, *Postn*, *Tnc*, *Pth1r*, *Sox9*, *Runx2*, *Sp7*, *Atf4*). *$p < 0.0001$, Wilcoxon rank sum test. Circle size: percentage of cells expressing a given gene in a given cluster (0–100%), Color density: expression level of a given gene. **e**, **f** CFU-F assay of Cxcl12$^{CE}$-tdTomato⁺ bone marrow cells, harvested from contralateral (CONT, left) and ablated (ABL, right) femurs at Day 7 (pulsed at Day-7). Scale bars: 5 mm. **g** Percentage of tdTomato⁺ colonies among total CFU-Fs. $n = 5$ mice per group. *$p < 0.05$, two-tailed, Mann–Whitney's $U$ test. Data are presented as mean ± s.d. Source data are provided as a Source Data file.

transformation of mature stromal cells. This is reminiscent of a model proposed in the adult liver and pancreas, both of which possess high regenerative capacity despite slow cell turnover; these organs primarily depend on mature cells for tissue maintenance in homeostasis and regeneration in response to injury[31]. We assume that mature stromal cells abundantly available in the milieu provide the primary source of cells executing bone repair under markedly slow cell turnover in the adult bone, in conjunction with a small number of resident osteoblast precursor cells. These highly malleable, long-living mature stromal cells

readily available throughout the lifespan represent an ideal cellular source for tissue regeneration. Further mechanistic studies are needed to exploit these cells better for future regenerative medicine.

## Methods

**Generation of Cxcl12-creER BAC transgenic mice.** *Cxcl12-iCreERT2-bGHpA* transgenic mice were generated by pronuclear injection of a modified BAC clone RP23-22F6 (Children's Hospital Oakland Research Institute, Oakland, CA) containing 74 kb upstream and 121 kb downstream genomic sequences of the Cxcl12 gene. Hybrid primers (Fw: ACTTTCACTCTCGGTCCACCTCGGTGTCCTC

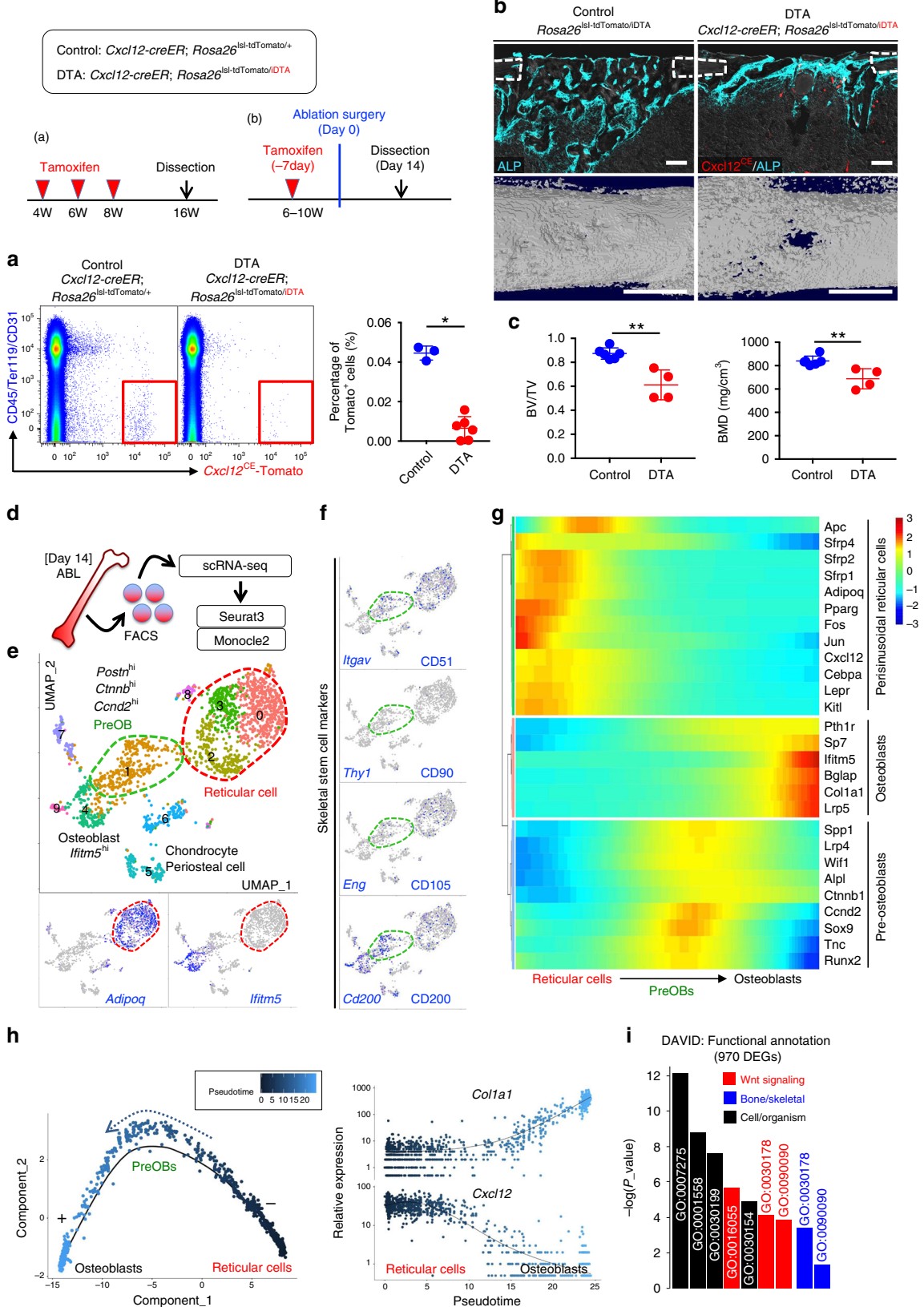

TTGCTGTCCAGCTCTGCAGCCTCCGGCGCGCCCTCCCGCCCACGCCACC ATGGTGCCCAAGAAGAAGAG, Rv: CGGGCCCGGCCGCGGATTGCACTCAC CGTCACTGATGCAGAGCGCGGCCAGCACCAGGGCCAGCACGGCGACG ACCTCTATACGAAGTTATAAGCTT) containing 77 bp homology arms specific to the first exon of the *Cxcl12* gene were used to polymerase chain reaction (PCR)-amplify the targeting construct from the plasmid containing the *iCreERT2-bGHpA*

sequence. The PCR product was gel-purified prior to electroporation into *Escherichia coli* strain SW105 containing RP23-22F6. Recombinants were selected on bacterial plates containing kanamycin and PCR-screened for homologous recombination events, followed by pulsed-field gel electrophoresis analysis (CHEF-DRIII, Bio-rad) to confirm no gross rearrangement of the targeted BAC clones. The sequence of the BAC-insert junction was further validated based on PCR. The

**Fig. 7 Cxcl12-creER$^+$ BMSCs transform into a skeletal stem cell-like state. a** Inducible partial cell ablation of *Cxcl12-creER$^+$* stromal cells using an inducible diphtheria toxin fragment A (DTA) allele. Flow cytometry-based verification of iDTA-mediated cell ablation of *Cxcl12-creER$^+$* stromal cells. $n = 3$ (tdT), $n = 6$ (DTA) mice. *$p < 0.05$, two-tailed, Mann–Whitney's *U* test. **b, c** DTA-mediated ablation of *Cxcl12-creER$^+$* cells during cortical bone regeneration. *Rosa26$^{lsl-tdTomato/iDTA}$* (Control) and *Cxcl12-creER; Rosa26$^{lsl-tdTomato/iDTA}$* (DTA) femur diaphyseal cortical bones, 14 days after cortical drill-hole surgery (pulsed at 7 days before surgery). **b** Cortical bone injured areas. Upper panels: alkaline phosphatase (ALP) staining. Gray: DIC. Scale bar: 200 μm. Lower panels: 3D-μCT. Scale bar: 1 mm. **c** Bone volume/tissue volume (BV/TV, left) and bone mineral density (BMD, right) of the injured area. $n = 6$ (Control), $n = 4$ (DTA) mice. **$p < 0.01$, two-tailed, Mann–Whitney's *U* test. **d–i** Single-cell RNA-seq analyses of lineage-traced *Cxcl12-creER$^+$* cells. Cxcl12$^{CE}$-tdTomato$^+$ cells were isolated from ablated (ABL) femurs at 14 days after marrow ablation. **e** UMAP-based visualization of major classes of mesenchymal Cxcl12$^{CE}$-tdTomato$^+$ cells (Clusters 0–9). Red dotted contour: reticular cells, green dotted contour: preosteoblasts. $n = 1825$ cells pooled from $n = 7$ mice. **f** Feature plots of skeletal stem cell marker genes. **g** Heatmap of representative pseudotime-dependent genes. **h** Pseudotime and single cell trajectory analysis by Monocle. **i** Most significant Gene Ontology (GO) terms (Biological Process) based on DAVID functional annotation of differentially expressed genes (DEGs). Black: general, red: Wnt signaling-related, blue: bone-related terms. All data are presented as mean ± s.d. Source data are provided as a Source Data file.

NeoR cassette was removed from correctly targeted BAC clones by L-arabinose induction of flpe recombinase, and lox sites (loxP and lox511) in the BAC plasmid were replaced with AmpR and NeoR cassettes by recombineering. The modified BAC clone was verified by pulsed-field gel electrophoresis analysis, and purified by NucleoBond Xtra BAC purification kit (Clontech). The BAC clone was linearized by BsiWI digest, and a 207 kb fragment containing *Cxcl12-iCreERT2-bGHpA* was separated by size-exclusion chromatography (Sepharose CL-4B). The fragment eluted from the column was collected in multiple fractions in polyamine micro-injection buffer to protect the DNA from shearing, followed by pulsed-field gel electrophoresis analysis. The concentration of the BAC construct was verified on Nanodrop 2000 spectrophotometer. The BAC clone was microinjected into B6SJLF1 fertilized eggs at a concentration of 0.25-0.5 ng/μl. Genomic DNA from potential founders were genotyped with PCR primers (320: TTTCACTCTCG GTCCACCTC, 321: GGATCTGCACACAGACAGGA), in which one primer is designed external to homology arms while another primer is designed internal to the transgene. Generic iCre primers (322: GTGCAAGCTGAACAACAGGA, 323: CCAGCATCCACATTCTCCTT) were also used. The founder mice were back-crossed with C57/BL6 mice at least for four generations before analysis. The three lines were established; Line274, Line277, and Line289. Line274 showed no recombination in bone marrow stromal cells when bred to an *R26R-tdTomato* allele and exposed to tamoxifen; tdTomato$^+$ cells were exclusively discovered in the articular cartilage and its surrounding tissues. Line277 showed extensive tamoxifen-independent recombination when bred to an *R26R-tdTomato* allele. Line289 showed no tamoxifen-independent recombination in the stromal cell compartment when bred to an *R26R-tdTomato allele*, and bone marrow stromal cells were marked in response to tamoxifen administration. Therefore, Line289 was used for this study.

**Mice.** *Cxcl12$^{GFP/+}$* [6] and *Osx-creER* [19] mice have been described previously. *Rosa26-CAG-loxP-stop-loxP-tdTomato* (Ai14: *R26R-tdTomato*, JAX007914), *Col1a1*(2.3 kb)-GFP (JAX013134), *osteocalcin (Ocn)*-GFP (JAX017469), *Ubc-creER* (JAX007001), *Dlx5-creER* (JAX010705), *Rosa26-SA-loxP-GFP-stop-loxP-DTA* (JAX006331), *Ctnnb*-floxed (JAX004152), *Apc*-floxed (JAX009045), and *Sox9*-floxed (JAX013106)[32] mice were acquired from the Jackson laboratory. *Runx2*-floxed mice[33] were provided from Dr. Cecilia Giachelli. All procedures were conducted in compliance with the Guidelines for the Care and Use of Laboratory Animals approved by the University of Michigan's Institutional Animal Care and Use Committee (IACUC), protocol 7681 (Ono) and 8779 (Kozloff). All mice were housed in a specific pathogen-free condition, and analyzed in a mixed background. For all breeding experiments, *creER* transgenes were maintained in male breeders to avoid spontaneous germline recombination. Mice were identified by micro-tattooing or ear tags. Tail biopsies of mice were lysed by a HotShot protocol (incubating the tail sample at 95 °C for 30 min in an alkaline lysis reagent followed by neutralization) and used for PCR-based genotyping (GoTaq Green Master Mix, Promega, and Nexus X2, Eppendorf). Perinatal mice were also genotyped fluor-escently (BLS miner's lamp) whenever possible. Mice were euthanized by over-dosage of carbon dioxide or decapitation under inhalation anesthesia in a drop jar (Fluriso, Isoflurane USP, VetOne).

**Tamoxifen.** Tamoxifen (Sigma T5648) was mixed with 100% ethanol until completely dissolved. Subsequently, a proper volume of sunflower seed oil (Sigma S5007) was added to the tamoxifen-ethanol mixture and rigorously mixed. The tamoxifen-ethanol-oil mixture was incubated at 60ºC in a chemical hood until the ethanol evaporated completely. The tamoxifen-oil mixture was stored at room temperature until use. Tamoxifen was injected to mice intraperitoneally using a 26-1/2-gauge needle (BD309597).

**High fat diet containing rosiglitazone.** Mice were fed with a high-fat diet containing 20 mg/kg rosiglitazone (D10010301, 60 kcal%, Research Diets) for 6 weeks.

**Surgery.** Mice were anesthetized through a nose cone, in which 1.5–2% isoflurane was constantly provided with oxygen through a vaporizer.

For drill-hole injury, left femurs were operated, while right femurs were untreated and used as an internal control. An incision was made in the lateral skin of the mid-femur. After splitting the muscle, the periosteum was separated to expose the femoral surface. A drill-hole injury was made onto the diaphysis by a Micro-Drill (Stoelting) with a 0.9 mm diameter stainless steel burr (Fine Science Tools). The periosteum of the surgical field was scraped off using a scalpel prior to the procedure.

For bone marrow ablation surgery, right femurs were operated, while left femurs were untreated and used as an internal control. After skin incision, cruciate ligaments of the knee were carefully separated using a dental excavator, and a hole was made in the intercondylar region of femurs using a 26-1/2-gauge needle (BD309597). The endodontic instruments (K-file, #35, 40, 45, and 50) (GC, Tokyo, Japan) were used in a stepwise manner to remove a cylindrical area of the marrow space. The surgical field was irrigated with saline, and the incision line was sutured.

For tibial complete fracture healing model, legs were aseptically prepared using Chlorhexidine scrub, once mice were anesthetized. A small incision was made on the anterior-medial surface of the proximal tibial metaphysis, exposing the knee joint and patellar tendon. Using a small gauge needle, an entry hole was made distally through the proximal tibial condyles, through which a narrow stainless steel wire was placed and extended through the intramedullary canal until contact at the distal end. The pin was trimmed flush with the proximal tibial condyles, and skin and soft tissue are closed with suture. While remaining under anesthesia, animals were positioned under a custom-built guillotine apparatus such that upon impact, a transverse fracture was made at the mid-distal tibial diaphysis. Tibias were gently stabilized with a small length of medical tape to provide support and biomechanical stability for immediate ambulation. Animals were subsequently recovered and returned to housing cages.

To evaluate cell proliferation, 5-ethynyl-2′-deoxyuridine (EdU) (Invitrogen A10044) dissolved in phosphate-buffered saline (PBS) was administered twice (6 and 3 h) before sacrifice at 2, 7, and 14 days after drill-hole injury (dose per injection: 200 μg).

**Histology and immunohistochemistry.** Samples were fixed in 4% paraf-ormaldehyde for a proper period, typically ranging from 3 h to overnight at 4 °C, then decalcified in 15% ethylenediaminetetraacetic acid for a proper period, typi-cally ranging from 3 h to 14 days. Decalcified samples were cryoprotected in 30% sucrose/PBS solutions and then in 30% sucrose/PBS:OCT (1:1) solutions, each at least overnight at 4 °C. Samples were embedded in an OCT compound (Tissue-Tek, Sakura) and cryosectioned at 14 μm using a cryostat (Leica CM1850) and adhered to positively charged glass slides (Fisherbrand ColorFrost Plus). Sections were postfixed in 4% paraformaldehyde for 15 min at room temperature. For immunostaining, sections were permeabilized with 0.25% TritonX/TBS for 30 min, blocked with 3% bovine serum albumin/TBST for 30 min, and incubated with rat anti-endomucin (Emcn) monoclonal antibody (1:100, Santa Cruz Biotechnology, sc65495), rat anti-CD31 monoclonal antibody (1:100, Bio-Rad, MCA2388), goat anti-leptin receptor (LepR) polyclonal antibody (1:100, R&D, AF497), goat anti-osteopontin (OPN) polyclonal antibody (1:500, R&D, AF808), goat anti-ALPL polyclonal antibody (1:100, R&D, AF2910), rabbit anti-Sox9 polyclonal antibody (1:500, EMD-Millipore, AB5535), rabbit anti-perilipin A/B polyclonal antibody (1:500, Sigma, P1873) or rabbit anti-PPAR gamma (PPARγ) monoclonal antibody (1:200, Invitrogen, MA5-14889) overnight at 4 °C, and subsequently with Alexa Fluor 633-conjugated goat anti-rat IgG (A21049), Alexa Fluor 647-conjugated goat anti-goat IgG (A21082), or Alexa Fluor 647-conjugated donkey anti-rabbit IgG (A31573) (1:400, Invitrogen) for 3 h at room temperature. For EdU assay, sections were incubated with Alexa Fluor 647-azide (Invitrogen, A10277) for 30 min at 43 °C using Click-iT Imaging Kit (Invitrogen, C10337). Sections were further incubated with DAPI (4′,6-diamidino-2-phenylindole, 5 μg/ml, Invitrogen D1306) to stain nuclei prior to imaging. For lipid staining, cryosections were immediately cover-slipped with LipidTOX Deep Red (1:200, Invitrogen, H34477)

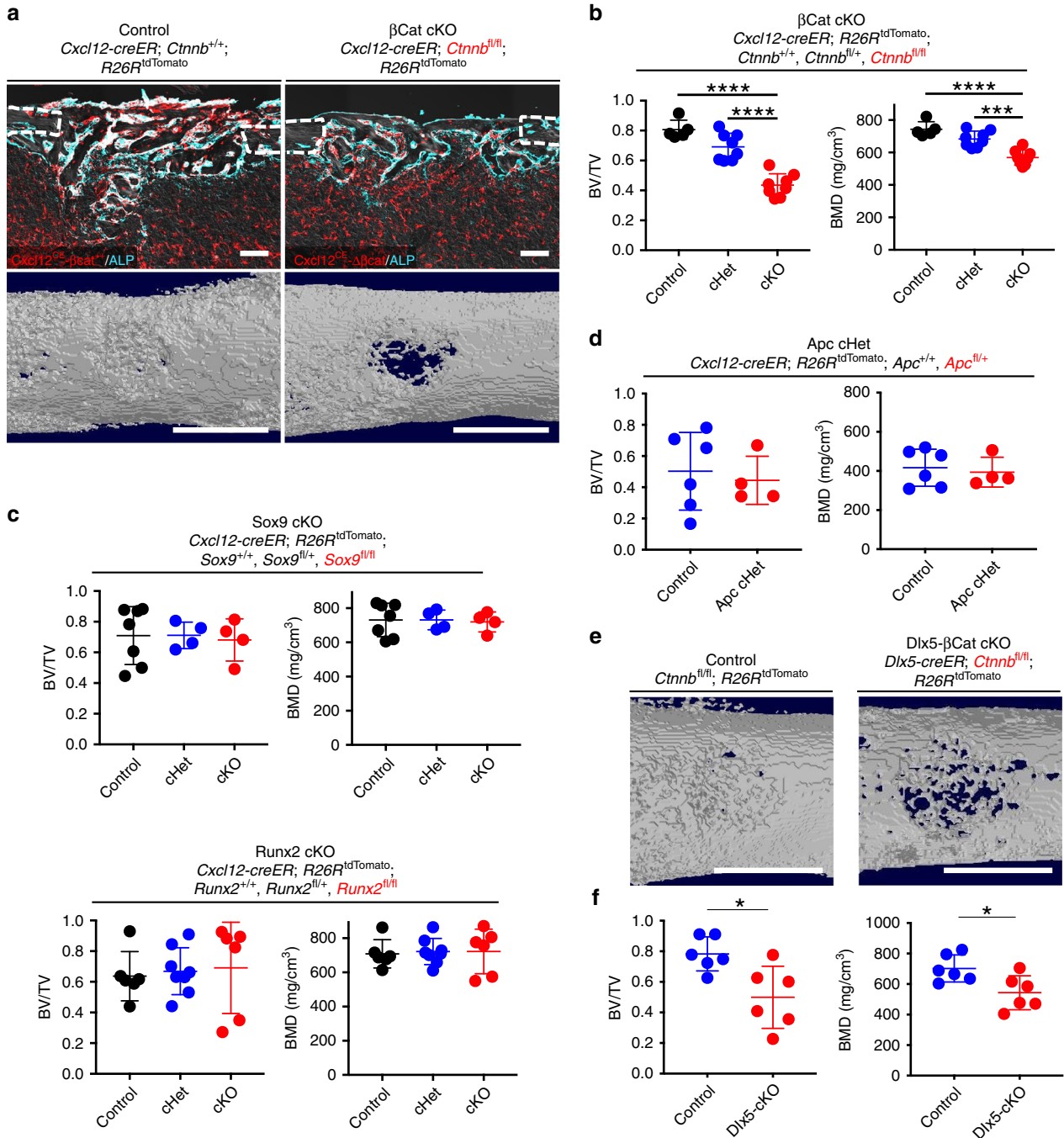

**Fig. 8 Wnt-mediated transformation orchestrates bone regeneration. a, b** Functional analysis of Wnt/β-Catenin signaling in *Cxcl12-creER*+ cells during cortical bone regeneration. *Cxcl12-creER; Ctnnb*+/+; *R26R*tdTomato (Control, left), *Cxcl12-creER; Ctnnb*fl/fl; *R26R*tdTomato (βcat cKO, right) femur diaphyseal cortical bones, 14 days after cortical drill-hole surgery (pulsed at 7 days before surgery). **a** Cortical bone injured areas. Upper panels: alkaline phosphatase (ALP) staining. Gray: DIC. Scale bar: 200 μm. Lower panels: 3D-μCT. Scale bar: 1 mm. **b** Bone volume/tissue volume (BV/TV, left) and bone mineral density (BMD, right) of the injured area. *n* = 4 (Control), *n* = 8 (*Cxcl12-creER; Ctnnb*fl/+; *R26R*tdTomato, βCat cHet) mice, *n* = 8 (βCat cKO) mice. **c** Functional analysis of Sox9 and Runx2 in *Cxcl12-creER*+ cells during cortical bone regeneration. *Cxcl12-creER; Sox9*+/+ or *Runx2*+/+;*R26R*tdTomato (Control, left), *Cxcl12-creER; Sox9*fl/+ or *Runx2*fl/+;*R26R*tdTomato (Sox9 cHet or Runx2 cHet, center) and *Cxcl12-creER; Sox9*fl/fl or *Runx2*fl/fl; *R26R*tdTomato (Sox9 cKO or Runx2 cKO, right) femur diaphyseal cortical bones, 14 days after cortical drill-hole surgery (pulsed at 7 days before surgery). BV/TV (left) and BMD (right) of the injured area. *n* = 7 (Sox9 Control), *n* = 4 (Sox9 cHet, cKO) mice. *n* = 6 (Runx2 Control, cKO) mice. *n* = 8 (Runx2 cHet) mice. **d** Functional analysis of Apc in *Cxcl12-creER*+ cells during cortical bone regeneration. *Cxcl12-creER; Apc*+/+; *R26R*tdTomato (Control, left), *Cxcl12-creER; Apc*fl/+; *R26R*tdTomato (Apc cHet, right) femur diaphyseal cortical bones, 7 days after cortical drill-hole surgery (pulsed at 7 days before surgery). BV/TV (left) and BMD (right) of the injured area. *n* = 6 (Control), *n* = 4 (Apc cHet) mice. **e, f** Functional analysis of Wnt/β-Catenin signaling in *Dlx5-creER*+ cells during cortical bone regeneration. *Ctnnb*fl/fl; *R26R*tdTomato (Control, left), *Dlx5-creER; Ctnnb*fl/fl; *R26R*tdTomato (Dlx5-βcat cKO, right) femur diaphyseal cortical bones, 14 days after cortical drill-hole surgery (pulsed at 7 days before surgery). **e** 3D-μCT. Lower panels: BV/TV (left) and BMD (right) of the injured area. Scale bar: 1 mm. *n* = 6 mice per group. ****p* < 0.0001, ***p* < 0.001, *p* < 0.05, two-tailed, one-way ANOVA followed by Tukey's post hoc test (**b**, **c**). Two-tailed, Mann–Whitney's *U* test (**d**, **f**). All data are presented as mean ± s.d. Source data are provided as a Source Data file.

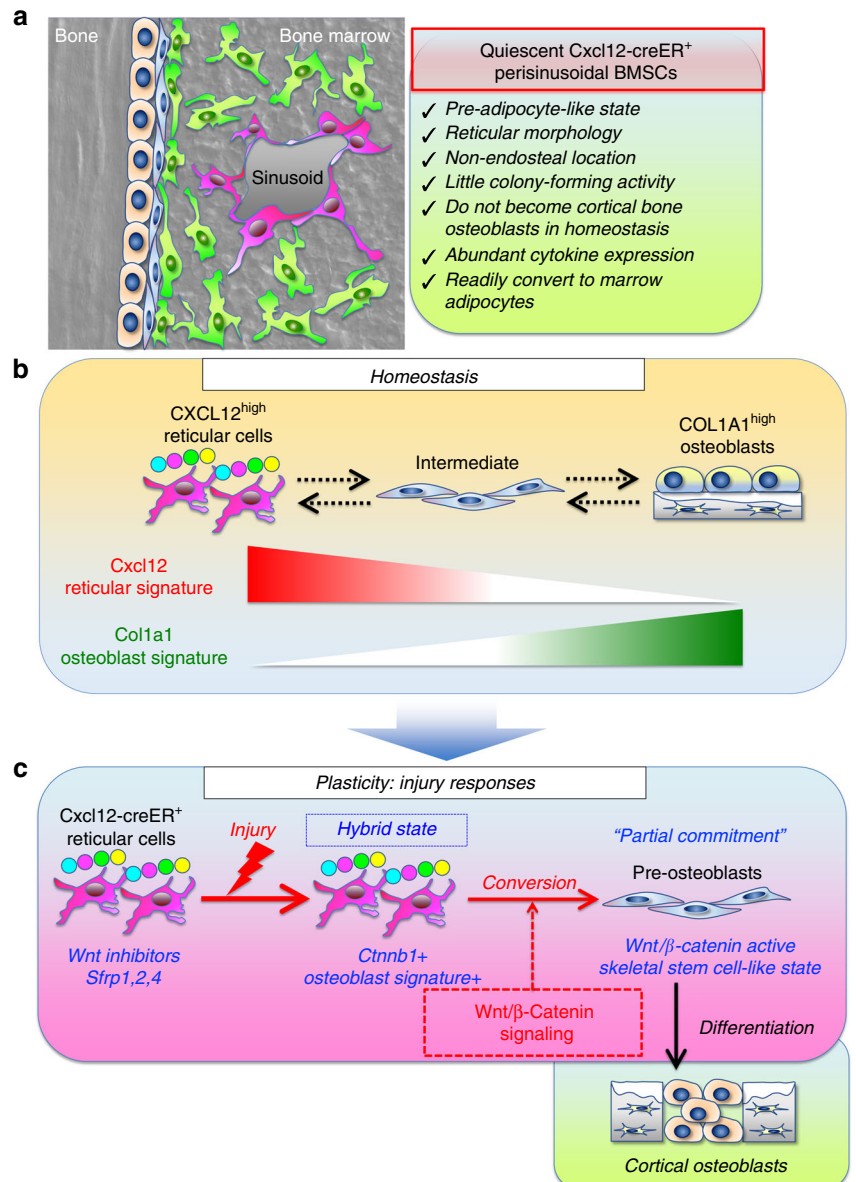

**Fig. 9 A Wnt-mediated conversion of stromal cell identity orchestrates skeletal regeneration. a** Quiescent CXCL12+ bone marrow stromal cells (BMSCs) in perisinusoidal space are marked by *Cxcl12-creER* upon tamoxifen injection. These cells are preferentially localized in a non-endosteal location of the marrow space, associated with a reticular morphology, abundant cytokine expression and little colony-forming activities. These cells do not become cortical bone osteoblasts in homeostasis, but can be readily converted to marrow adipocytes. **b** A proposed diagram of the bone marrow stromal cell lineage. In homeostasis, CXCL12high reticular cells and COL1A1high osteoblasts represent two opposite differentiated states of the bone marrow stromal cell lineage. The transition between these two cell types is continuous, wherein intermediate-state cells represent a precursor cell population. **c** Wnt-mediated cellular plasticity of CXCL12+ reticular cells during injury responses. In homeostasis, Cxcl12-creER+ reticular cells are in a Wnt-inhibitory state by expressing potent Wnt inhibitors, such as *Sfrp1*, *Sfrp2*, and *Sfrp4*. During injury responses, Cxcl12-creER + reticular cells are brought into a reticular-osteoblast hybrid state by injury-induced cues, associated with upregulation of osteoblast-signatures and a critical canonical Wnt component, β-catenin. Subsequently, Wnt/β-catenin signaling induces cell identity conversion of Cxcl12-creER+ reticular cells into pre-osteoblasts, which are featured by a skeletal stem cell-like state with partial commitment to the osteoblast lineage. These cells can be recruited to cortical bone defect and functionally contribute to skeletal regeneration.

and 5 μg/ml DAPI solution and incubated for 2 h at 43 °C. Stained samples were mounted in TBS with No. 1.5 coverslips (Fisher).

**Imaging.** Images were captured by an automated inverted fluorescence microscope with a structured illumination system (Zeiss Axio Observer Z1 with ApoTome.2 system) and Zen 2 (blue edition) software. The filter settings used were: FL Filter Set 34 (Ex. 390/22, Em. 460/50 nm), Set 38 HE (Ex. 470/40, Em. 525/50 nm), Set 43 HE (Ex. 550/25, Em. 605/70 nm), Set 50 (Ex. 640/30, Em. 690/50 nm) and Set 63 HE (Ex. 572/25, Em. 629/62 nm). The objectives used were: Fluar 2.5×/0.12, EC Plan-Neofluar 5×/0.16, Plan-Apochromat 10×/0.45, EC Plan-Neofluar 20×/0.50, EC Plan-Neofluar 40×/0.75, Plan-Apochromat 63×/1.40. Images were typically tile-

scanned with a motorized stage, Z-stacked and reconstructed by a maximum intensity projection function. Differential interference contrast was used for objectives higher than 10×. Representative images of at least three independent biological samples are shown in the figures. Quantification of cells on sections was performed using NIH Image J software.

**Cell dissociation.** Soft tissues and epiphyses were carefully removed from dissected femurs. After removing distal epiphyseal growth plates and cutting off proximal ends, femurs were cut roughly and incubated with 2 Wunsch units of Liberase™ (Sigma/Roche 5401127001) and 1 mg of Pronase (Sigma/Roche 10165921001) in 2 ml Ca2+, Mg2+-free Hank's Balanced Salt Solution (HBSS, Sigma H6648) at

37 °C for 60 min. on a shaking incubator (Thermomixer$^R$, Eppendorf). After cell dissociation, cells were mechanically triturated using an 18-gauge needle with a 1 ml Luer-Lok syringe (BD) and a pestle with a mortar (Coors Tek), and subsequently filtered through a 70 µm cell strainer (BD) into a 50 ml tube on ice to prepare single cell suspension. These steps were repeated for five times, and dissociated cells were collected in the same tube. Cells were pelleted and resuspended in an appropriate medium for subsequent purposes. For cell culture experiments, cells were resuspended in 10 ml culture medium and counted on a hemocytometer.

**Flow cytometry**. Dissociated cells were stained by standard protocols with the following antibodies (1:500, Invitrogen/eBioscience). eFluor450-conjugated CD31 (390, endothelial/platelet, 48-0311-82), CD45 (30F-11, hematopoietic, 48-0451-82), Ter119 (TER-119, erythrocytes, 48-5921-82), allophycocyanin (APC)-conjugated CD31 (390, endothelial/platelet, 17-0311-82), CD45 (30F-11, hematopoietic, 17-0451-82), Ter119 (TER-119, erythrocytes, 17-5921-82). For LepR staining, goat anti-LepR polyclonal antibody (1:200, R&D, AF497) and Alexa Fluor 647-conjugated donkey anti-goat IgG (1:400, Invitrogen, A21082) were used. For intracellular CXCL12 staining, cells were fixed and permeabilized using Intracellular Fixation & Permeabilization Buffer (eBioscience), and stained for APC-conjugated CXCL12 (1:100, R&D, IC350A). For intracellular SCF staining, cells were fixed and permeabilized using Intracellular Fixation & Permeabilization Buffer (eBioscience), and incubated with rabbit anti-SCF polyclonal antibody (1:50, Bioss, bs-0545R), and subsequently with Alexa Fluor 647-conjugated donkey anti-goat IgG (1:400, Invitrogen, A21082).

Cell cycle analyses were performed using DAPI (5 µg/ml). For EdU assays, two doses of EdU were injected shortly before analysis, at 6 and 3 h prior to sacrifice, or serial doses of EdU were injected for 3 days (3 times a day, total 9 injections). These pulsed mice were analyzed by histology or flow cytometry using the Click-iT EdU Alexa Flour 647 Assay kit (Invitrogen, C10634).

Flow cytometry analysis was performed using a four-laser BD LSR Fortessa (Ex. 405/488/561/640 nm) and FACSDiva software. Acquired raw data were further analyzed on FlowJo software (TreeStar). Representative plots of at least three independent biological samples are shown in the figures.

**Single-cell RNA-seq analysis of FACS-isolated cells**. Cell sorting was performed using a four-laser BD FACS Aria III (Ex.407/488/561/640 nm) or a six-laser Sony Synergy SY3200 (Ex.350/405/488/561/594/685 nm) high-speed cell sorter with a 100 µm nozzle. GFP$^+$ cells or tdTomato$^+$ cells were directly sorted into ice-cold Dulbecco's phosphate-buffered saline (DPBS)/10% fetal bovine serum (FBS), pelleted by centrifugation and resuspended in 10 µl DPBS/1% FBS. Cell numbers were quantified by Countless II automated Cell Counter (ThermoFisher) before loading onto the Chromium Single Cell 3′ v2 or v3 microfluidics chip (10× Genomics Inc., Pleasanton, CA). cDNA libraries were sequenced by Illumina HiSeq 4000 using two lanes and 50 cycle paired-end read, generating a total of ~770 million reads, or NovaSeq 6000. The sequencing data was first pre-processed using the 10× Genomics software Cell Ranger. For alignment purposes, we generated and used a custom genome fasta and index file by including the sequences of *eGFP* and *tdTomato-WPRE* to the mouse genome (mm10). Further downstream analysis steps were performed using the Seurat and Monocle R package. We filtered out cells with less than 500 genes per cell and with more than 20% mitochondrial read content for v2, or with less than 1000 genes per cell and with more than 15% mitochondrial read content for v3. The downstream analysis steps include normalization, identification of highly variable genes across the single cells, scaling based on number of UMI, dimensionality reduction (PCA, CCA, and UMAP), unsupervised clustering, and the discovery of differentially expressed cell-type specific markers. Differential gene expression to identify cell-type specific genes was performed using the nonparametric Wilcoxon rank sum test.

**Colony-forming assay**. Nucleated bone marrow cells were plated into tissue culture 6-well plates (BD Falcon) at a density of <10$^5$ cells/cm$^2$, and cultured in low-glucose DMEM with GlutaMAX supplement (Gibco 10567022) and 10% mesenchymal stem cell-qualified FBS (Gibco 12662029) containing penicillin–streptomycin (Sigma P0781) for 10–14 days. Cell cultures were maintained at 37 °C in a 5% CO$_2$ incubator. Representative images of at least three independent biological samples are shown in the figures.

**Subcloning and in vitro trilineage differentiation**. Colonies marked by tdTomato were individually subcloned. A tile-scanned virtual reference of tdTomato$^+$ colonies and cloning cylinders (Bel-Art) were used to isolate an individual colony. Single cell-derived clones of tdTomato$^+$ cells were cultured in a basal medium described above at 37 °C with 5% CO$_2$ with exchanges into fresh media every 3–4 days. Cells at Passage 4–7 were used for trilineage differentiation. To induce chondrocyte differentiation, cells were transferred into a non-tissue culture-coated V-bottom 96-well plate (Corning 3896). The plate was centrifuged at 150$g$ for 5 min at room temperature to pellet the cells, and the supernatant was carefully aspirated. StemPro Chondrogenesis medium (ThermoFisher A1007101) was gently added, and the plate was centrifuged at 150$g$ for 5 min at room temperature to

pellet the cells. The pellet was cultured in the differentiation medium with exchanges into fresh media every 3–4 days for up to 3 weeks, each time with centrifugation at 150$g$ for 5 min at room temperature to repellet the cells. Cell pellets were fixed with 70% ethanol for 5 min. and stained for Alcian-Blue Staining Solution (Millipore TMS-010-C) for 30 min. To induce osteoblast differentiation, cells were plated on a 48-well plate and cultured with αMEM with GlutaMAX supplement (Gibco 32571036) with 10% FBS containing 1 µM dexamethasone (Sigma D4902), 10 mM β-glycerophosphate (Sigma G9422) and 50 µg/ml ascorbic acid (Sigma A5960) with exchanges into fresh media every 3–4 days for up to 3 weeks. Cells were fixed with 4% paraformaldehyde for 5 min and stained for 2% Alizarin Red S (Sigma A5533) for 30 min. To induce adipocyte differentiation, cells were plated on a 48-well plate and cultured with αMEM with GlutaMAX supplement with 10% FBS containing 1 µM dexamethasone (Sigma D4902), 0.5 mM IBMX (3-Isobutyl-1-methylxanthine, Sigma I5879) and 1 µg/ml insulin (Sigma I6634) with exchanges into fresh media every 3–4 days for up to 2 weeks. Cells were stained with LipidTOX Green (Invitrogen H34475, 1:200 in basal medium) for 3 h at 37 °C, or fixed with 4% paraformaldehyde for 5 min and stained for Oil Red O (Sigma O0625).

**Subcutaneous transplantation of cultured cells**. NOD scid gamma (NSG) (JAX005557) mice were used as recipients for cell transplantation. Mice were anesthetized through a nose cone, in which 1.5–2% isoflurane is constantly provided with oxygen through a vaporizer. Fur was removed with clippers and depilatory cream in a clean area. Matrigel Matrix (Corning) was mixed with 10$^6$–10$^7$ cells (cultured *Cxcl12-creER$^+$* stromal cells, Passage 10 or higher), and loaded into an insulin syringe (BD Lo-Dose U-100 Insulin Syringes, BD329461). The cell–matrix mixture was injected subcutaneously into a designated spot in the back skin or the calvaria.

**Three-dimensional micro-computed tomography analysis**. Femurs were dissected and scanned by micro-computed tomography (microCT) (eXplore Locus SP, GE Healthcare Pre-Clinical Imaging, London, ON, Canada). Scans were reconstructed at 18 µm isotropic voxel size and analyzed using Microview v2.2 (GE Healthcare, London, Ontario). To analyze the healing of the 0.9 mm drill hole defect in the diaphyseal femur, images were re-oriented to align the direction of drilling with the $x$-axis. A 0.7 mm diameter cylindrical region of interest (ROI) was centered immediately inside the cortex, and in the cortical gap with a fixed thickness of 0.2 mm. This ROI typically spanned the cortical thickness from the endosteal-to-periosteal surface. Regenerated sites were assessed for bone volume fraction using a threshold of 2300 Hounsfield Units (HU). A common threshold was calculated across all samples by using the auto-threshold tool of microView for each specimen. 3D images were reconstructed using CT-Analyzer (Bruker microCT).

**Statistical analysis**. Results are presented as mean values ± S.D. Statistical evaluation was conducted using the Mann–Whitney's $U$ test or one-way ANOVA. A $p$ value <0.05 was considered significant. No statistical method was used to predetermine sample size. Sample size was determined on the basis of previous literature and our previous experience to give sufficient standard deviations of the mean so as not to miss a biologically important difference between groups. The experiments were not randomized. All of the available mice of the desired genotypes were used for experiments. The investigators were not blinded during experiments and outcome assessment. One femur from each mouse was arbitrarily chosen for histological analysis. Genotypes were not particularly highlighted during quantification.

**Reporting summary**. Further information on research design is available in the Nature Research Reporting Summary linked to this article.

## Data availability

The datasets generated during and/or analyzed during the current study are available from the corresponding author on reasonable request. The single-cell RNA-seq data presented herein have been deposited in the National Center for Biotechnology Information (NCBI)'s Gene Expression Omnibus (GEO), and are accessible through GEO Series accession numbers GSE136970, GSE136973, and GSE136979. The source data underlying all Figures and Supplementary Figures are provided as a Source Data file.

## Code availability

The custom codes utilized for single cell RNA-seq analysis of this study are publicly available from the following websites: Seurat 3.0 R package (https://satijalab.org/seurat/) and Monocle 2 R package (http://cole-trapnell-lab.github.io/monocle-release/).

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

## Acknowledgements

This research was supported by grants from National Institute of Health (R01DE026666 to N.O., R03DE027421 to W.O., P30AR069620 to Michigan Integrated Musculoskeletal Core Center), the Uehara Memorial Foundation Research Fellowship and the Japan Society for the Promotion of Science Overseas Research Fellowship to Y.M. We thank C. Giachelli (University of Washington) for *Runx2* floxed mice, M. Pihalja and K. Saiya-Cork of the University of Michigan Flow Cytometry Core, T. Saunders, G. Gavrilina and W. Fillipak of the University of Michigan Transgenic Animal Model Core, T. Tamsen, J. Opp of the University of Michigan Advanced Genomics Core, R. Tagett of the University of Michigan Bioinformatics Core, and C. Stephan, B. Nolan and A. Clark of University of Michigan Medical School Orthopedic Research Laboratories and T. Lau for supporting this study.

## Author contributions

Y.M. and N.O. conceived the project and wrote the paper. Y.M., M.N., K.M., K.K., N.T., S.A.H., and N.O. performed the experiment. T.N. and D.L. provided the mice. J.D.W., T.N., D.L., and W.O. critiqued the paper.

## Competing interests

The authors declare no competing interests.
