## [Peer Review File · Nature Communications]

Reviewers' comments:

Reviewer #1 (Remarks to the Author):

This is really an outstanding paper with substantial implications for the field of skeletal stem cell biology. The observation that a bone injury can induce Wnt pathway activation that converts a subset of LepR⁺ cells from being poised to undergo adipogenic differentiation (under steady-state conditions) to undergoing osteogenic differentiation to repair the bone injury is fundamentally important. The greatest compliment I can pay to a paper is that it changes the way I think about the biology – this paper changes the way I think about the biology. Matsushita et al. identified a Cxcl12-creER⁺ subset of perisinusoidal LepR⁺ cells, which contribute to trabecular bone and fat under steady state, but begin giving rise to cortical bone during regeneration after a bone injury. Although Cxcl12 is expressed by nearly all LepR⁺ cells, the Cxcl12-creER allele recombines in only around 30% of LepR⁺ cells, apparently the ones that express the highest levels of Cxcl12. The recombined LepR⁺ cells are mostly (>95%) perisinusoidal, making this strain a valuable tool to functionally assess this perisinusoidal subset of LepR⁺ cells. The paper is technically very well done and the data are convincing.

1. A minor point that can be addressed by changes in the text is that the authors should be careful to acknowledge that their conclusions apply only to the 30% of perisinusoidal Cxcl12⁺ stromal cells that are labelled by Cxcl12-CreER. In the current manuscript, the conclusions are sometimes stated in a way that implies that all perisinusoidal Cxcl12⁺ cells have the same properties. For example, in the abstract the authors write "...revealed that quiescent CXCL12⁺ perisinusoidal BMSCs differentiated into cortical bone osteoblasts solely during regeneration." However, their Cxcl12-creER allele recombined in only around 30% of CXCL12⁺ perisinusoidal BMSCs, and other fate mapping data show that Lepr-cre labelled cells (which are uniformly Cxcl12⁺) are fated to give rise to cortical bone (Cell Stem Cell 15:154). An important future question raised by the current manuscript is whether the LepR⁺ cells that contribute to cortical bone are among the 70% of perisinusoidal cells that are not recombined by Cxcl12-creER, or by periarteriolar, or metaphyseal cells. Answering this question is clearly outside of the scope of the current manuscript, but the conclusions of this manuscript should be stated in a way that doesn't pre-suppose an answer. For example, the sentence in the abstract could be revised to "...revealed that quiescent CXCL12-creER⁺ perisinusoidal BMSCs differentiated into cortical bone osteoblasts solely during regeneration." Other text in the body of the manuscript could be similarly revised.

2. How long did the authors administer EdU to the mice in the experiments in which they concluded that Cxcl12-creER⁺ cells are entirely EdU negative (Fig S5C). LepR⁺ cells have been shown to be mostly quiescent under steady-state conditions, with less than 10% of them incorporating BrdU over a two-week period (Cell Stem Cell 15:154). It would be surprising if the Cxcl12-creER⁺ cells are entirely quiescent. The authors should clarify this, because a low frequency of dividing cells is biologically quite different from no dividing cells. They should state how long they administered EdU, and if only for a very short period of time (e.g. hours) they should perform a longer EdU administration (e.g. 3 days) to more carefully test whether the cells are completely quiescent.

3. The authors should add data to Figure 1I also comparing the levels of Scf (kitl) among the subpopulations of bone marrow stromal cells shown in that figure (I presume they already have these data). This is interesting because the LepR⁺ cells are a required source of both Cxcl12 and SCF for adult HSC maintenance (Nature 481:457, Nature 495:227 and Nature 495:231) and the evidence in Figure 1I of differences in Cxcl12 expression level among these cells is interesting. Do the same cells also exhibit differences in SCF expression level?

Bo Shen and Sean Morrison

Reviewer #2 (Remarks to the Author):

This is an interesting study using lineage tracing to track cells near blood vessels that express C-X-C motif chemokine ligand 12 during osteogenesis and repair processes. State of the art techniques are utilized.

There is some general over interpretation of data, especially around the single cell data – there are caveats to the interpretation, and these should be more clearly stated. For instance, this data is focused on CXCL12+ cells, and without knowing how other osteoblast progenitor populations respond, it is difficult to make statements too broadly about the implications of this data. Expression patterns can also be skewed by the numbers of genes assessed as well.

It is unclear how single femur ablation studies were performed. More details about this as well as appropriate control data is required to assess the reliability of the data.

It is well known that beta-catenin is important in differentiation to osteoblasts in fracture repair (e.g. Chen Y, et al . Beta-catenin signaling plays a disparate role in different phases of fracture repair: implications for therapy to improve bone healing. PLoS Med. 2007), and previous lineage tracing works shows that multiple cells can participate in becoming osteoblasts (Liu R, Birke O, et al. BMC Musculoskelet Disord. 2011). Are CXCL12+ cells more or less sensitive to beta-catenin than other fracture repair sources? Is the balance of beta-catenin important (will high levels inhibit differentiation to osteoblasts as well)?

In Fig 2e, data on the other growth plate should be shown as well (the distal growth plate). The geometry of the images give the impression that the stained cells may be just adjacent to the distal growth plate, but it is difficult to tell as the other growth plate is missing from the images.

What is the proportion of mesenchymal cells in the healing process from the CXCL12+ progenitor? This is important in light of the possibility that multiple cell sources may participate in fracture repair, and it will be important to determine what proportion of cells come from this cell source.

The interpretation of the rainbow studies are confusing – given the labeling process it is not clear why one would expect anything other than multi colored cells. The authors may want to remove this from the manuscript.

Reviewer #3 (Remarks to the Author):

The manuscript by Matsushita et al. details the application of a Cxcl12-creER mouse model to study the transition of bone marrow stromal cells (BMSC) into osteogenic cells which contribute to bone regeneration following insult. In this review I will focus primarily on single-cell RNA-seq component of the manuscript as was specifically requested by the editor.

The first set of single-cell data is displayed in figure 1J – which shows 10X Chromium analysis of FACS sorted Cxcl12-GFP+ cells. A total of 2135 cells are detected in this analysis (S2C) of which 3 clusters (351 cells) are in clusters in which the eGFP transcript is detected. These 351 cells are selected for further analysis and these Cxcl12-GFP+ cells cluster into four groups - including two clusters of mesenchymal (Cluster 0,1 – pre osteoblast and adipocyte) and endothelial (Cluster 2,3).

Firstly, it is unclear from the manuscript if the 2135 cells presented in figure S2C are derived from

sorting of GFP⁺ cells. If so, it should be explained why the vast majority of the cells do not express the eGFP transcript. There appears to be considerable heterogeneity between the 2135 cells, and so it should be discussed what they are, and if based on GFP⁺ sort, why they are not positive for the transcript.

Secondly, within the 351 GFP⁺ cells, it is suggested that cluster 1 is “enriched for tdTomato” but it appears that is based on detection of the transcript in dozen or so cells. No scale is presented in this figure to indicate the number of tdTomato transcripts detected per cell, it may be that expression of this transcript is low, and thus ruling its expression out in other cells from other clusters may not be possible, as the level of expression of the transcript is approaching the limit of detection of the 10x genomics approach.

Generally, there is no detail on the distribution of the number of transcripts detected per cell (just >500 genes per cell) which can be helpful in making a technical appraisal of the data. Some supplementary QC of the data would be useful.

The second set of single-cell data is presented in Figure 3 (and S7), in which tdTomato⁺ cells have been isolated by FACS and again processed using the 10X Chromium platform, 6539 cells of which 2,569 cells are in clusters which express tdTomato transcripts. Again, it is not clear why only a small proportion of the cells are positive for the sorted marker and what the other detected clusters are. Here, the experimental model is to ablate the bone marrow and compare with a control sample (the contralateral femur from the same mouse). The authors here indicate that this experiment identifies a) expansion of osteoblast populations from 7.3% in the control sample to 31.1% in the ablated sample and b) differential expression of osteoblast marker genes in the reticular cell clusters.

For a) as this is based on a single experiment from a single mouse, I would caution that it remains unknown what the accuracy of platforms like 10x is when it comes to cell quantification, especially at such low cell numbers – clusters 2 and 5 are comprised of ~200 cells (estimating by eye) so there are relatively few events on which this statistic is based. A replicate run (or two) would be essential to confirm that this frequency is reproducible.

Within Cluster 5, Ifitm5 is mentioned as a marker of terminal differentiation of osteoblasts. Only a subset of the osteoblast population appear Ifitm5 positive in Figure S7, and this subset seems, arguably, to be evenly comprised of control and ablated cells.

There is considerable risk in making quantitative observations about relative cell abundances from unproven technology such as droplet based single cell RNA-seq. Thus replicate measurements (or orthogonal validation) of cell counts such as this – especially with such rare cells – are essential. This is validated in part through colony forming assays, but here I am commenting specifically on the single cell data.

For b) it does indeed seem that there are significant changes in gene expression within clusters 0, 1 and 2, with osteoblast markers increased in the ablated cells in all clusters. It is not clear why this does not drive the cells to cluster separately per condition. It would be of interest to compare the expression levels of these markers between the clusters to understand how the expression levels of these markers compare between all of the clusters/conditions.

The third set of single cell data is another ablation experiment in which again tdTomato cells were sorted and analysed by 10x genomics. Again, only a fraction of the cells (1825/9474 cells) expressed the tdTomato transcript. These cells show similar clustering to the previous experiment (here there is not control femur). Pseudotime ordering of these cells was used to confirm a linear “trajectory” from reticular cell to pre-osteoblast to osteoblast, and that Wnt signalling components displayed regulated expression along this trajectory.

Overall I think the single cell approaches used in the paper are appropriate, and serve the purpose of hypothesis generation in the manuscript. However the authors are drawing conclusions from the single-cell data that are beyond the technical limitations of the approach – from single replicate experiments it is simply not possible to quantify reliably changes in cell abundance and even measurements of differential expression between samples can be confounded by batch effects. I am also not clear on why so few of the sorted cells express the selection marker (perhaps I have missed

something) and why there is no discussion of the 'other cells' that make up most of the single-cell data in the paper. As I said, this is right at the technical limits of what 10X genomics can do in terms of gene detection and cell counting, and the authors should critically appraise this in any revision of the manuscript.

Bone marrow stromal cells – CXCL12+ (cre-ER mice)

Injury makes CXcl12+ cells convert identity into skeletal stem cell-like state and upregulation of osteoblast genes including Wnt. B-catenin down = drop in bone marrow regeneration. MEDIATED BY CANONICAL WNT?

tG characterisation -not mature osteoblasts (Col1a1 negative) subset of LepR+ quiescent single cell

Cxcl12-GFP+ cells were heterogeneous and clustered into four groups, including two clusters of mesenchymal (Cluster 0,1 – pre osteoblast and adipocyte) and endothelial (Cluster 2,3)

Cluster 1 enriched for tdTomato – a dozen or so cells in how many? Would have been better to sort tdTomato

Cluster 0 were more proliferative – what does this mean in the context of a static measurement.

Few tomato + cells in cluster 1

Tomato cells are rich in adipocyte related genes and cytokines. In vitro give adipocytes chondrocytes and osteoblasts. Can self renew

Quantitative nature of sc-RNA for cell population counts, appears to be a single experiment with no replication and a very small number of cells in the population enriched for tdTomato

Ablated/control

Saying something is not expressed is not possible with 10x data – was not detected.

1. Comments for Author

What are the major claims of the paper?

Are they novel and will they be of interest to others in the community and the wider field?

If the conclusions are not original, it would be helpful if you could provide relevant references. Is the work convincing, and if not, what further evidence would be required to strengthen the conclusions?

On a more subjective note, do you feel that the paper will influence thinking in the field? Please feel free to raise any further questions and concerns about the paper.

We would also be grateful if you could comment on the appropriateness and validity of any statistical

analysis, as well the ability of a researcher to reproduce the work, given the level of detail provided.

To increase the transparency and openness of the reviewing process, we do support our reviewers signing their reports to authors if the reviewers feel comfortable doing so. If, however, you prefer to send an anonymized report we will continue to respect and maintain your anonymity. Referee reports, whether signed or not, are subsequently shared with the other reviewers.

Response to Reviewer #1

Dear Bo and Sean,

Thank you very much for your very supportive comments. We believe that our manuscript will add substantial insight into distinct functionality of subsets of LepR⁺ BMSCs. We also believe that our *Cxcl12-creER* line will become a useful tool for the relevant field, particularly for skeletal stem cell biologists and hematopoietic stem cell nicheologists.

In the revised manuscript, we have provided additional evidence for quiescence of *Cxcl12-creER*⁺ perisinusoidal stromal cells as well as SCF expression levels in these cells, following your suggestions.

Sincerely,

Yuki Matsushita & Noriaki Ono

1. <Reviewer>

This is really an outstanding paper with substantial implications for the field of skeletal stem cell biology. The observation that a bone injury can induce Wnt pathway activation that converts a subset of LepR⁺ cells from being poised to undergo adipogenic differentiation (under steady-state conditions) to undergoing osteogenic differentiation to repair the bone injury is fundamentally important. The greatest compliment I can pay to a paper is that it changes the way I think about the biology – this paper changes the way I think about the biology. Matsushita et al. identified a *Cxcl12-creER*⁺ subset of perisinusoidal LepR⁺ cells, which contribute to trabecular bone and fat under steady state, but begin giving rise to cortical bone during regeneration after a bone injury. Although *Cxcl12* is expressed by nearly all LepR⁺ cells, the *Cxcl12-creER* allele recombines in only around 30% of LepR⁺ cells, apparently the ones that express the highest levels of *Cxcl12*. The recombined LepR⁺ cells are mostly (>95%) perisinusoidal, making this strain a valuable tool to functionally assess this perisinusoidal subset of LepR⁺ cells. The paper is technically very well done and the data are convincing.

<Response>

Thank you very much for your very supportive comments on our study!

2. <Reviewer>

1. A minor point that can be addressed by changes in the text is that the authors should be careful to acknowledge that their conclusions apply only to the 30% of perisinusoidal Cxcl12+ stromal cells that are labelled by Cxcl12-CreER. In the current manuscript, the conclusions are sometimes stated in a way that implies that all perisinusoidal Cxcl12+ cells have the same properties. For example, in the abstract the authors write "...revealed that quiescent CXCL12+ perisinusoidal BMSCs differentiated into cortical bone osteoblasts solely during regeneration." However, their Cxcl12-creER allele recombined in only around 30% of CXCL12+ perisinusoidal BMSCs, and other fate mapping data show that LepR-cre labelled cells (which are uniformly Cxcl12+) are fated to give rise to cortical bone (Cell Stem Cell 15:154). An important future question raised by the current manuscript is whether the LepR+ cells that contribute to cortical bone are among the 70% of perisinusoidal cells that are not recombined by Cxcl12-creER, or by periarteriolar, or metaphyseal cells. Answering this question is clearly outside of the scope of the current manuscript, but the conclusions of this manuscript should be stated in a way that doesn't pre-suppose an answer. For example, the sentence in the abstract could be revised to "...revealed that quiescent CXCL12-creER+ perisinusoidal BMSCs differentiated into cortical bone osteoblasts solely during regeneration." Other text in the body of the manuscript could be similarly revised.

<Response>

We have revised the term 'CXCL12+' cells into 'Cxcl12-creER+' cells throughout the manuscript, to clarify the specific subset of CXCL12+ cells that can be marked by *Cxcl12-creER* upon tamoxifen injection.

We completely agree with you that LepR+ cells that contribute to cortical bone osteoblasts in physiological conditions are within 70% of perisinusoidal cells that are not marked by *Cxcl12-creER*. Interestingly, a recent single cell RNA-seq study (Nature, 569:222-228) shows that *LepR-cre-tdTomato*+ BMSCs are comprised of four subpopulations (i.e. Mgp^{high}, Lpl^{high}, Wif1^{high} and Spp1^{high}), revealing their cellular heterogeneity. Our single cell RNA-seq data in **Figure 1J,K** shows that *Cxcl12-creER* can preferentially mark a subpopulation with abundant adipocyte-related gene expression, which may correspond to Lpl^{high} population in the above-mentioned paper. The identities of the other three subpopulations are unknown; these cells may represent periarteriolar cells or other stromal cells, as you pointed out.

We believe that functional heterogeneity of LepR+CXCL12+ cells is an important topic in this field, and we would like to pursue this in our future studies.

The following sentence was added to the Discussion:

LepR+CXCL12+ cells that contribute to cortical bone osteoblasts in physiological conditions are within the majority of perisinusoidal cells that are not marked by Cxcl12-creER.

3. <Reviewer>

2. How long did the authors administer EdU to the mice in the experiments in which they concluded that Cxcl12-creER+ cells are entirely EdU negative (Fig S5C). LepR+ cells have been shown to be mostly

quiescent under steady-state conditions, with less than 10% of them incorporating BrdU over a two-week period (Cell Stem Cell 15:154). It would be surprising if the *Cxcl12*-creER⁺ cells are entirely quiescent. The authors should clarify this, because a low frequency of dividing cells is biologically quite different from no dividing cells. They should state how long they administered EdU, and if only for a very short period of time (e.g. hours) they should perform a longer EdU administration (e.g. 3 days) to more carefully test whether the cells are completely quiescent.

<Response>

For the experiments described in **Fig.S5C**, we injected two doses of EdU shortly before analysis, at 6 and 3 hours prior to sacrifice. This point has been clarified in the revised legend of **Figure S5C**.

Following your suggestions, we have now performed a longer EdU administration to strengthen our claim on relative quiescence of *Cxcl12*-creER⁺ cells. We treated *Cxcl12*^{GFP/+}; *Cxcl12*-creER; *R26R*^{tdTomato} mice first with tamoxifen at P21, and subsequently injected serial doses of EdU for 3 days (3 times a day, total 9 injections) between P25 and P28. These pulsed mice were analyzed by flow cytometry at P28 using the Click-iT EdU Alexa Flour 647 Assay kit (**##1**).

Shown below is the percentage of EdU-Alexa647⁺ cells among CD45/Ter119/CD31^{neg}GFP^{high}tdTomato^{neg} (left) and CD45/Ter119/CD31^{neg}GFP^{high}tdTomato⁺ (right) cells.

Figure R1-1 (new Figure S1C)

As shown in **Figure R1-1**, 5.1±3.1% and 1.3±0.7% of *Cxcl12*-GFP^{high}tdTomato^{neg} cells and *Cxcl12*-GFP^{high}tdTomato⁺ cells were positive for EdU, respectively. The difference was statistically significant. This data has been incorporated as new **Figure S1C**.

The finding supports the statement that *Cxcl12*-creER can mark a relatively quiescent subset of CXCL12⁺ BMSCs. However, this data also supports the statement that *Cxcl12*-creER⁺ cells are not completely quiescent, as a small fraction of *Cxcl12*-creER⁺ cells can indeed incorporate EdU when it is repetitively injected, unlike when it is injected only twice shortly before analysis. In addition, as shown in **Figure 1H**, our cell cycle analysis demonstrates that a small fraction of *Cxcl12*-GFP^{high}tdTomato⁺ cells (i.e.

3.6±1.5%) were in S/G2/M, whereas a larger fraction of tdTomato^{neg} cells were in S/G2/M (i.e. 7.8±3.1% of Cxcl12-GFP^{high}tdTomato^{neg} cells, 32.6±4.4% of Cxcl12-GFP^{neg} cells).

Therefore, you are right that these Cxcl12-creER⁺ cells are not entirely quiescent; rather, they are characterized as cells relatively more quiescent than other stromal cells. This point has been emphasized in the revised manuscript.

4. <Reviewer>

The authors should add data to Figure 1I also comparing the levels of Scf (kitl) among the subpopulations of bone marrow stromal cells shown in that figure (I presume they already have these data). This is interesting because the LepR⁺ cells are a required source of both Cxcl12 and SCF for adult HSC maintenance (Nature 481:457, Nature 495:227 and Nature 495:231) and the evidence in Figure 1I of differences in Cxcl12 expression level among these cells is interesting. Do the same cells also exhibit differences in SCF expression level?

<Response>

Following your suggestions, we have analyzed intracellular SCF protein levels of CD45/Ter119/CD31^{neg} populations of bone marrow cells isolated from P28 *Cxcl12*^{GFP/+}; *Cxcl12-creER*; *R26R*^{tdTomato} mice (tamoxifen pulse at P21), using a commercially available anti-SCF antibody (Bioss SCF polyclonal antibody, bs-0545R). Shown below is MFI (median fluorescence intensity) of intracellular SCF protein of four subpopulations of CD45/Ter119/CD31^{neg} cells e.g. GFP^{neg}, GFP^{low}, GFP^{high}Tom^{neg} and GFP^{high}Tom⁺ cells.

Figure R1-2 (new Figure S1D)

As shown in **Figure R1-2**, Cxcl12-GFP^{high}tdTomato^{neg} and Cxcl12-GFP^{high}tdTomato⁺ cells exhibited significantly higher levels of intracellular SCF expression than that of Cxcl12-GFP^{neg} cells, as expected. Interestingly, there was no statistical significance between Cxcl12-GFP^{high}tdTomato^{neg} and Cxcl12-GFP^{high}tdTomato⁺ cells, indicating that these subsets of CXCL12⁺ BMSCs do not exhibit a difference in terms of SCF expression. This data indicates that Cxcl12-GFP^{high}tdTomato⁺ cells that express CXCL12 most abundantly do not necessarily express SCF most abundantly. This data has been incorporated as new **Figure S1D**.

We believe that the difference between CXCL12 and SCF is interesting in light of overlapping but biologically distinct functions of these two proteins in HSC maintenance, as your group has convincingly demonstrated for years. We assume that the most significant source of SCF might be a subset of LepR⁺CXCL12⁺ bone marrow stromal cells that are distinct from Cxcl12-creER⁺ cells. We think this is an important agenda that needs to be addressed through further experimentation.

Response to Reviewer #2

Thank you very much for your constructive comments and critiques.

1. <Reviewer>

This is an interesting study using lineage tracing to track cells near blood vessels that express C-X-C motif chemokine ligand 12 during osteogenesis and repair processes. State of the art techniques are utilized.

<Response>

Thank you very much for your supportive comments and suggestions on our study.

2. <Reviewer>

There is some general over interpretation of data, especially around the single cell data – there are caveats to the interpretation, and these should be more clearly stated. For instance, this data is focused on CXCL12+ cells, and without knowing how other osteoblast progenitor populations respond, it is difficult to make statements too broadly about the implications of this data. Expression patterns can also be skewed by the numbers of genes assessed as well.

<Response>

We agree with the reviewer that defining the relationship of Cxcl12-creER⁺ cells and other precursor populations is important. Particularly, how other osteoprogenitor populations can respond to injury would be essential to interpreting our data. The data we show in **Figure 2N** demonstrate that 39.7±1.9% of osteocytes in the regenerated portion of the cortical bone were derived from Cxcl12-creER⁺ cells. This data also suggests that ~60% of osteocytes in the regenerated portion might be derived from other progenitor populations.

To address the reviewer's question, we have performed additional experiments. We have taken advantage of an *osterix* (*Osx*)-*creER* line, which can mark osteoblast precursor cells and mature osteoblasts, and evaluated how these can contribute to the cortical bone healing process (**##2**). For these experiments, *Osx-creER; R26R^{tdTomato}* mice were pulsed at 6 – 10 weeks of age with the cortical bone injury being induced at one week later, and were analyzed after 8 weeks of injury. We enumerated the percentage of tdTomato⁺ osteocytes within the regenerated portion of the cortical bone to determine contribution of Osx⁺ osteoblast precursor cells to cortical bone regeneration.

Figure R2-1 (revised Figure 2N,M, new Figure S5H)

As shown in **Figure R2-1**, $\text{Osx}^{\text{CE}}\text{-tdTomato}^+$ cells moderately contributed to osteocytes in the regenerated portion of the cortical bone ($\text{Osx}^{\text{CE}}\text{-tdTomato}^+$ osteocytes: $12.3\pm 4.9\%$ of total osteocytes). The contribution of $\text{Osx}^{\text{CE}}\text{-tdTomato}$ cells to osteocytes in the regenerated portion was significantly less than that of $\text{Cxcl12}^{\text{CE}}\text{-tdTomato}^+$ cells. Therefore, Osx-creER^+ cells might be one of other progenitor populations that contribute to approximately 60% of osteocytes in the regenerated portion that are not derived from Cxcl12-creER^+ BMSCs. This supports the reviewer's point that multiple types of mesenchymal cells can participate in bone repair.

Regarding the reviewer's concern on the interpretation of single cell RNA-seq data, please see our responses to the **Reviewer #3**. We have carefully revised our statement to acknowledge the technical limitations associated with droplet-based single RNA-seq experiments in the revised manuscript.

The following paragraph was added to the Results, under the second section:

Multiple types of mesenchymal cells can participate in bone repair by becoming osteoblasts (Liu R et al., BMC Musculoskelet Disord, 2011). To address whether other osteoblast precursor cells can participate in cortical bone regeneration, we further took advantage of an osterix (Osx)-creER line. We performed the surgery at one week after Osx-creER; $R26R^{\text{tdTomato}}$ mice were pulsed at 6 – 10 weeks of age, and analyzed these mice after 8 weeks of injury. $\text{Osx}^{\text{CE}}\text{-tdTomato}^+$ cells moderately contributed to osteocytes in the regenerated portion of the cortical bone ($\text{Osx}^{\text{CE}}\text{-tdTomato}^+$ osteocytes: $12.3\pm 4.9\%$, Fig.2N, Fig.S5H,I). These data suggest that Osx-creER^+ cells might be one of other progenitor populations that contribute to approximately 60% of osteocytes in the regenerated portion that are not derived from Cxcl12-creER^+ BMSCs.

3. <Reviewer>

It is unclear how single femur ablation studies were performed. More details about this as well as appropriate control data is required to assess the reliability of the data.

<Response>

We would like to apologize for lack of explanation regarding our bone marrow ablation model in the original manuscript. We have now added more detailed explanations about our bone marrow ablation model in the revised manuscript. We have now included a dedicated supplementary figure (**new Figure S7**) to demonstrate the reliability of this bone marrow ablation procedure (**##5**).

Our femoral bone marrow ablation model utilizes a dental instrument K-file, which is clinically used for endodontic procedures. This procedure mechanically removes a cylindrical area (0.5mm diameter) of the bone marrow tissue. We have previously reported this procedure to induce direct differentiation of marrow stromal cells into osteoblasts within the marrow cavity, without involving the periosteum (J Cell Physiol. 2012, 227:408-15). We performed the surgery on the right femur to mechanically ablate a defined area of bone marrow. The left femur was untreated and used as an internal control. We have added the intraoperative pictures showing the step-by-step procedures in **new Figure S7A,B**.

Bone marrow ablation using an endodontic instrument is a highly reproducible model that induces highly sequential and predictable steps of osteogenesis within the marrow cavity. In **Figure S7C**, *Cxcl12-creER*; *R26R^{tdTomato}* mice carrying *Col1a1(2.3kb)-GFP* were pulsed at 6 – 10 weeks of age, and the bone marrow ablation surgery was performed at one week later.

Figure R2-2 (new Figure S7)

As shown in **Figure R2-2**, *Cxcl12^{CE}-tdTomato⁺* cells were robustly recruited to the ablated area as early as after 3 days of marrow ablation (**Figure S7D**). After 7 days of marrow ablation, *Cxcl12^{CE}-tdTomato⁺* cells started to differentiate into *Col1a1(2.3kb)-GFP⁺* osteoblasts within the marrow space, which was not observed in the contralateral control femur (**Figure 3B**). After 14 days of marrow ablation, *Cxcl12^{CE}-tdTomato⁺* cells extensively differentiated into *Col1(2.3kb)-GFP⁺* osteoblasts of newly formed woven bones within the marrow cavity (**Figure S7E,G**, contralateral control side shown in **Figure S7F**). The numbers of both total *Cxcl12^{CE}-tdTomato⁺* cells and differentiated *Col1(2.3kb)-GFP⁺Cxcl12^{CE}-tdTomato⁺* cells were significantly in the ablated femur compared to those of the contralateral control femur (**Figure S7H**), indicating that *Cxcl12-creER⁺* reticular cells can robustly proliferate and differentiate into osteoblasts in response to marrow ablation. In addition, these cells became chondrocytes in response to injury; distinct cartilaginous tissues composed of *Sox9⁺tdTomato⁺* cells were occasionally observed within the ablated area immediately beneath the growth plate (**Figure S7I**, 3/6 ablated bones, 50%).

Therefore, our bone marrow ablation model is a highly reproducible model that allows us to examine sequences of osteoblast differentiation *in vivo*. More specifically, we can examine the sequence at two distinct stages i.e. early events of osteoblast differentiation at 7 days of marrow ablation as well as complete events of osteoblast differentiation at 14 days of marrow ablation.

The following sentences were added to the Results, under the third section:

As an injury model, we made use of a femoral bone marrow ablation model that mechanically removes a defined area of bone marrow using an endodontic instrument (Ono et al, J Cell Physiol, 2012). This procedure induces direct differentiation of marrow stromal cells within the marrow cavity without involving the periosteum (see Fig.S7). The right femur underwent the surgery, while the left femur was untreated and used as an internal control.

4. <Reviewer>

It is well known that beta-catenin is important in differentiation to osteoblasts in fracture repair (e.g. Chen Y, et al. Beta-catenin signaling plays a disparate role in different phases of fracture repair: implications for therapy to improve bone healing. PLoS Med. 2007), and previous lineage tracing works shows that multiple cells can participate in becoming osteoblasts (Liu R, Birke O, et al. BMC Musculoskelet Disord. 2011). Are CXCL12+ cells more or less sensitive to beta-catenin than other fracture repair sources? Is the balance of beta-catenin important (will high levels inhibit differentiation to osteoblasts as well)?

<Response>

First, we have added these two references in the revised manuscript.

Second, to answer the reviewer's question whether CXCL12⁺ cells are more or less sensitive to β -catenin than other fracture repair sources, we have performed additional *Ctnnb* deletion experiments using another *creER* line that marks an osteoblast precursor population. *Dlx5* is a marker for early cells of the osteoblast lineage. We therefore have taken advantage of a *Dlx5-creER* line to evaluate how β -catenin deficiency in these osteoblast precursor cells can affect cortical bone regeneration. We conditionally deleted *Ctnnb* in *Dlx5-creER*⁺ osteoblast precursor cells during cortical bone regeneration, using *Ctnnb*^{*fl/fl*}; *R26R*^{*tdTomato*} (Control) and *Dlx5-creER*; *Ctnnb*^{*fl/fl*}; *R26R*^{*tdTomato*} (*Dlx5*- β Cat cKO) littermates. We performed the cortical drill-hole surgery at one week after these mice were pulsed with tamoxifen.

Figure R2-3 (new Figure S10A)

As shown in **Figure R2-3**, after 14 days of cortical bone injury, bone volume and bone mineral density of the injured cortical area were significantly reduced in Dlx5-βCat cKO mice compared with Control mice (**Figure S10A**). Therefore, Dlx5-creER⁺ osteoblast precursor cells are equally as sensitive to β-catenin as Cxcl12-creER⁺ BMSCs.

Third, to answer the reviewer's question whether high levels of β-catenin can inhibit osteoblast differentiation, we have induced haploinsufficiency of *adenomatous polyposis coli* (*Apc*), which is a critical component of the β-catenin degradation complex, using its floxed allele in Cxcl12-creER⁺ BMSCs during cortical bone regeneration. *Cxcl12-creER; Apc^{+/+}; R26R^{tdTomato}* (Control) and *Cxcl12-creER; Apc^{fl/+}; R26R^{tdTomato}* (*Apc* cHet) littermates received a tamoxifen pulse at 6 – 10 weeks of age, and underwent surgery at one week after.

Figure R2-4 (new Figure S10B)

As shown in **Figure R2-4**, after 14 days of cortical bone injury, there was no significant change in bone volume and bone mineral density of the injured cortical area in Apc cHet mice (**Figure S10B**). Therefore, a high level of β-catenin in Cxcl12-creER⁺ BMSCs does not inhibit their osteoblast differentiation during cortical bone regeneration.

The following paragraphs were added to the Results, under the third section:

Canonical Wnt signaling plays a disparate role in different phases of bone fracture repair (Chen Y et al, *PLOS Med*, 2007). To determine whether Cxcl12-creER⁺ cells are more or less sensitive to β-catenin

than other fracture repair sources, we conditionally deleted *Ctnnb* in osteoblast precursor cells during cortical regeneration, using *Ctnnb^{fl/fl}; R26R^{tdTomato}* (Control) and *Dlx5-creER; Ctnnb^{fl/fl}; R26R^{tdTomato}* (*Dlx5-βCat* cKO) littermates. After 14 days of cortical bone injury, bone volume and bone mineral density of the injured cortical area were significantly reduced in *Dlx5-βCat* cKO mice compared with Control mice (Fig.S10A). Therefore, *Cxcl12-creER⁺* BMSCs are equally as sensitive to β -catenin as *Dlx5-creER⁺* osteoblast precursor cells.

Further, to address whether high levels of β -catenin can inhibit osteoblast differentiation of *Cxcl12-creER⁺* BMSCs during cortical bone regeneration, we conditionally induced haploinsufficiency of adenomatous polyposis coli (*Apc*), which is a critical component of the β -catenin degradation complex, during cortical bone regeneration using *Cxcl12-creER; Apc^{+/+}; R26R^{tdTomato}* (Control) and *Cxcl12-creER; Apc^{fl/+}; R26R^{tdTomato}* (*Apc* cHet) littermates. After 14 days of cortical bone injury, there was no significant change in bone volume and bone mineral density of the injured cortical area in *Apc* cHet mice (Fig.S10B). Therefore, a high level of β -catenin in *Cxcl12-creER⁺* BMSCs does not inhibit their osteoblast differentiation during cortical bone regeneration.

The following sentence was added to the Discussion:

CXCL12⁺ BMSCs are equally as sensitive to deficiency in canonical Wnt signaling as other osteoblast precursor cells.

5. <Reviewer>

In Fig 2e, data on the other growth plate should be shown as well (the distal growth plate). The geometry of the images give the impression that the stained cells may be just adjacent to the distal growth plate, but it is difficult to tell as the other growth plate is missing from the images.

<Response>

Following the reviewer's suggestion, we have now added images with intact proximal growth plates in the revised **Figure S4**.

Figure R2-3 (revised Figure S4H,I)

As shown in **Figure R2-3**, Cxcl12^{CE}-tdTomato⁺ cells were adjacent to the proximal growth plate, in a distinct manner to that of the distal growth plate, reflecting the fact that the most active bone elongation and enlargement occur at the distal side in femurs (Stern, et al. PLOS Biol. 2015). This indeed supports the concept stated in the original manuscript that Cxcl12-creER⁺ cells are dormant and remain in the original domain of the marrow space in growing bones, without producing new stromal cells in the newly formed domain of the marrow space in the metaphyseal region resulting from long-bone growth.

6. <Reviewer>

What is the proportion of mesenchymal cells in the healing process from the CXCL12+ progenitor? This is important in light of the possibility that multiple cell sources may participate in fracture repair, and it will be important to determine what proportion of cells come from this cell source.

<Response>

As shown in the revised **Figure 2N**, Cxcl12-creER⁺ BMSCs contribute to 39.7±1.9% of osteocytes within the regenerated cortical bone after 8 weeks of injury, whereas Osx-creER⁺ osteoblast precursor cells contribute to 12.3±4.9% of osteocytes at the same time point. This difference was statistically significant. As mentioned above, approximately 60% of osteocytes in the regenerated portion might be derived from other progenitor populations, supporting the reviewer's point that multiple type of mesenchymal cells can participate in bone repair.

The following underlined sentence was added in the Discussion:

We assume that mature stromal cells abundantly available in the milieu provide the primary source of cells executing bone repair under markedly slow cell turnover in the adult bone, in conjunction with a small number of resident osteoblast precursor cells.

7. <Reviewer>

The interpretation of the rainbow studies are confusing – given the labeling process it is not clear why one would expect anything other than multi colored cells. The authors may want to remove this from the manuscript.

<Response>

Thank you very much for this suggestion. We have now removed these data in the revised manuscript.

Response to Reviewer #3

Thank you very much for your constructive comments and critiques.

1. <Reviewer>

The manuscript by Matsushita et al. details the application of a *Cxcl12*-creER mouse model to study the transition of bone marrow stromal cells (BMSC) into osteogenic cells which contribute to bone regeneration following insult. In this review I will focus primarily on single-cell RNA-seq component of the manuscript as was specifically requested by the editor.

<Response>

Thank you very much for your very helpful comments and suggestions for our study.

2. <Reviewer>

The first set of single-cell data is displayed in figure 1J – which shows 10X Chromium analysis of FACS sorted *Cxcl12*-GFP+ cells. A total of 2135 cells are detected in this analysis (S2C) of which 3 clusters (351 cells) are in clusters in which the eGFP transcript is detected. These 351 cells are selected for further analysis and these *Cxcl12*-GFP+ cells cluster into four groups - including two clusters of mesenchymal (Cluster 0,1 – pre osteoblast and adipocyte) and endothelial (Cluster 2,3).

Firstly, it is unclear from the manuscript if the 2135 cells presented in figure S2C are derived from sorting of GFP+ cells. If so, it should be explained why the vast majority of the cells do not express the eGFP transcript. There appears to be considerable heterogeneity between the 2135 cells, and so it should be discussed what they are, and if based on GFP+ sort, why they are not positive for the transcript.

<Response>

We would like to apologize for lack of clarification and omission of technical details in our original manuscript. The single cells presented in **Figure 1J,K** as well as **Figure S2C** are derived from the cells that were gated as GFP+ on the cell sorter, among cells obtained from *Cxcl12*-GFP+ bone marrow. We have now clarified the technical details in the revised manuscript (##5).

Following the reviewer's request for biological replicates in the later section, we have repeated this single cell RNA-seq experiment twice, each for independent biological sample (Mouse #1 and Mouse #2) (##4). The composite of these two new runs for two biological samples are now shown in the revised **Figure 1J,K**. We decided to replace the original single cell RNA-seq data with these new data, because the original data was from our very first 10X Chromium experiment in early 2018. Since then, we have performed more than 20 experiments for 10X Chromium, the protocol has been fully established in our lab.

We consistently use a widely utilized sorter (BD FACS AriaIII) with a 100µm nozzle (20 psi) and low-speed sorting (<10,000 events/second) to avoid damage to the cells and contamination of unwanted cells. Despite this, we consistently find at least some degree of contamination of cells not expressing eGFP transcript, particularly CD45+ (*Ptprc*+) hematopoietic cells (myeloid cells and lymphocytes) and erythroid cells. This phenomenon is particularly pronounced with FACS-sorting of bone marrow stromal cells in adult bones. We did not see such contamination in our growth plate chondrocyte single cell RNA-seq dataset, as

we published recently in JBMR (J Bone Miner Res. 2019 Mar 19. doi: 10.1002/jbmr.3719). Therefore, we believe this issue is specific to bone marrow stromal cells.

We believe that the contamination likely occurred during FACS-sorting. Shown below are representative images of enzymatically dissociated Cxcl12-GFP⁺ BMSCs (green) interrogated under the ImageStream imaging cytometer. Cells were also stained with CD45-APC (red) to detect intertwining hematopoietic cells (**##3**).

Figure R3-1

[Redacted]

As shown in **Figure R3-1**, about half of Cxcl12-GFP⁺ cells appeared to be conjugated with CD45-APC⁺ cells. We hypothesize that these cellular doublets become dissociated when they are loaded onto the oil droplet in the 10X Genomics 3' assay chip.

We believe it is extremely important to filter out transgene-negative (e.g. *GFP* or *tdTomato*-negative) cell clusters bioinformatically from the final two-dimensional plots. Our results suggest that this subtle technical effect occurs in many studies of bone marrow stromal cells.

For example, Baryawno et al recently published a single cell RNA-seq dataset of bone marrow stromal cells (Cell, 177:1-18). They isolated stromal cells by FACS-sorting live cells that are negative for hematopoietic cell markers, and subsequently analyzed 30,543 cells. Despite negative gating for hematopoietic cells during cell sorting, they reported that 9,647 of 30,543 cells (31.6%) showed typical expression of hematopoietic cell markers. This number is very similar to our new replicates shown in **Figure 1J,K**, in which 26.2% and 16.3% of sorted cells based on Cxcl12-GFP⁺ did not express *eGFP* transcripts. Overall, these findings indicate that hematopoietic cell contamination might be a common issue during isolation of bone marrow stromal cells based on cell sorting.

We have reported our findings concerning intertwined *eGFP*-positive and *eGFP*-negative cells in the revised manuscript.

The following underlined clauses were added to the Results, under the first section:

We further defined the identity of *Cxcl12*-creER⁺ stromal cells by a single cell RNA-seq analysis. To this end, we interrogated the profile of fluorescently sorted single cells gated on a GFP^{high} fraction isolated from *Cxcl12*^{GFP/+}; *Cxcl12*-creER; R26R^{tdTomato} bone marrow at P28 after a tamoxifen pulse at P21 (Mouse #1: 2,749 cells, Mouse #2: 4,577 cells, total 5,884 cells). Initial analysis revealed that a substantial fraction of cells (Mouse #1: 721/2,749 cells, 26.2%, Mouse #2: 747/4,577 cells, 16.3%) belonged to the clusters without detectable eGFP expression; these eGFP^{neg} clusters included myeloid cells, lymphocytes and erythroid cells (Fig.S2C). eGFP was exclusively expressed by cells that abundantly expressed *Cxcl12* (Fig.S2C).

3. <Reviewer>

Secondly, within the 351 GFP+ cells, it is suggested that cluster 1 is “enriched for tdTomato” but it appears that is based on detection of the transcript in dozen or so cells. No scale is presented in this figure to indicate the number of tdTomato transcripts detected per cell, it may be that expression of this transcript is low, and thus ruling its expression out in other cells from other clusters may not be possible, as the level of expression of the transcript is approaching the limit of detection of the 10x genomics approach.

<Response>

In the new single cell RNA-seq data that we present in the revised **Figure 1J,K**, *tdTomato* is identified as a cell-type specific marker for Cluster 0. We have revised the text accordingly. In addition, we have now included the violin plot for *tdTomato* in the revised manuscript in the revised **Figure 1J,K**, as well as in **Figure S2C**. These data are also shown below in **Figure R3-2**.

Figure R3-2 (revised Figure 1J)

As shown in **Figure R3-2**, cells expressing *tdTomato* could be clearly distinguished from those not expressing *tdTomato*, particularly in Cluster 0 and 2 based on the violin plot. Therefore, *tdTomato*⁺ cells could be observed as a distinct group of cells in these clusters.

As mentioned above, we have now repeated this single cell RNA-seq experiment twice with an updated and more sensitive 10X Genomic Chromium Single Cell 3' Reagent with v3 Chemistry. This has significantly improved the number of genes and UMIs detected per cell, therefore improving overall representation of *tdTomato* expression.

4. <Reviewer>

Generally, there is no detail on the distribution of the number of transcripts detected per cell (just >500 genes per cell) which can be helpful in making a technical appraisal of the data. Some supplementary QC of the data would be useful.

<Response>

Following the reviewer's suggestion, we have now included standard QC plots for each dataset in the revised manuscript, in **Figure S2C**, **Figure S8A** and **Figure S9D**. Of note, there appears to be a large difference between v2 Chemistry and v3 Chemistry in terms of the number of genes (feature) and UMIs (counts) detected, with v3 significantly more sensitive than v2.

5. <Reviewer>

The second set of single-cell data is presented in Figure 3 (and S7), in which tdTomato+ cells have been isolated by FACS and again processed using the 10X Chromium platform, 6539 cells of which 2,569 cells are in clusters which express tdTomato transcripts.

Again, it is not clear why only a small proportion of the cells are positive for the sorted marker and what the other detected clusters are.

<Response>

As we mentioned above, even if we sorted bone marrow stromal cells by FACS with a gate on a tdTomato⁺ fraction, we consistently found at least some degree of contamination of hematopoietic cells not expressing *tdTomato* transcript. As shown above by the ImageStream analysis, it appears that a large number of CXCL12⁺ cells can conjugate with hematopoietic cells.

Hematopoietic cell contamination might be a common issue during isolation of bone marrow stromal cells based on cell sorting. We believe that such contamination likely occurred during FACS-sorting. We also believe it is extremely important to filter out transgene-negative (e.g. *tdTomato*-negative) cell clusters bioinformatically from the final two-dimensional plots. Cells in other detected clusters without *tdTomato* transcript include CD45⁺ (*Ptprc*⁺) hematopoietic cells (e.g. myeloid cells and lymphocytes), and red blood cells.

We have clarified this issue regarding contamination of tdTomato-negative cells in the revised manuscript.

The following sentences were added to the Results, under the first section:

Initial analysis revealed that a substantial fraction of cells belonged to the clusters without detectable tdTomato expression; these tdTomato^{neg} clusters included myeloid cells, lymphocytes and erythroid cells (Fig.S8A). tdTomato was predominantly expressed by mesenchymal cells expressing Cxcl12 or Col1a1, but also by a small number of endothelial cells.

6. <Reviewer>

Here, the experimental model is to ablate the bone marrow and compare with a control sample (the contralateral femur from the same mouse). The authors here indicate that this experiment identifies a) expansion of osteoblast populations from 7.3% in the control sample to 31.1% in the ablated sample and b) differential expression of osteoblast marker genes in the reticular cell clusters. For a) as this is based on a single experiment from a single mouse, I would caution that it remains unknown what the accuracy of platforms like 10x is when it comes to cell quantification, especially at such low cell numbers – clusters 2 and 5 are comprised of ~200 cells (estimating by eye) so there are relatively few events on which this statistic is based. A replicate run (or two) would be essential to confirm that this frequency is reproducible.

<Response>

Following the reviewer's important suggestion, we have performed a replicate run of this particular single cell RNA-seq experiment. As shown in revised **Figure S8A**, we have integrated two datasets (Control and Ablated) from the second biological replicate bioinformatically, as we did for the first biological replicate shown in the original and revised manuscript.

Figure R3-3 (revised Figure 3CD-E & Figure S8B)

As shown in **Figure R3-3**, the second biological replicate (shown on the right) experiment showed a similar pattern to that of the first biological replicate (shown on the left). First, bone marrow ablation

appeared to induce a shift in cell populations more toward pre-osteoblasts and osteoblasts expressing *Alpl* and *Col1a1* (yellow arrowheads) in both biological replicates. Second, bone marrow ablation appeared to induce upregulation of osteoblast-signature genes. Therefore, we believe that our original findings on cell populations and gene expressions have been largely reproduced in the second replicate run.

However, we agree with the reviewer that this technology and the current bioinformatics approach may not be sensitive or reliable enough to make a conclusion on cell quantification. We therefore eliminated the quantification from the revised manuscript, and changed our statements accordingly.

Finally, we would like to point out that, in each single cell RNA-seq experiments, we pooled bone marrow cells from multiple mice, not from a single mouse.

The following sentence was added to the Results, under the third section:

Analysis of the second biological replicate showed a pattern similar to the first replicate (Fig.S8B).

7. <Reviewer>

Within Cluster 5, *Ifitm5* is mentioned as a marker of terminal differentiation of osteoblasts. Only a subset of the osteoblast population appear *Ifitm5* positive in Figure S7, and this subset seems, arguably, to be evenly comprised of control and ablated cells.

<Response>

We agree with the reviewer that only a subset of the osteoblast population within the cluster 4 (within the new revised integrated space) expresses *Ifitm5*, a well-established marker for terminally differentiated osteoblasts. We have revised this sentence as following to clarify this important point:

Original: cells in Cluster 5 expressed a terminally differentiated osteoblast marker *Ifitm5*.

Revised: a terminally differentiated osteoblast marker *Ifitm5* was exclusively expressed by cells in Cluster 4.

8. <Reviewer>

There is considerable risk in making quantitative observations about relative cell abundances from unproven technology such as droplet based single cell RNA-seq. Thus replicate measurements (or orthogonal validation) of cell counts such as this – especially with such rare cells – are essential. This is validate in part through colony forming assays, but here I am commenting specifically on the single cell data.

<Response>

We believe that the second biological replicate shown above supports the biological reproducibility of the change in cell populations and gene expressions. However, following the reviewer's important suggestion, we have now eliminated the quantitative descriptions about relative cell abundances in the revised manuscript.

9. <Reviewer>

For b) it does indeed seem that there are significant changes in gene expression within clusters 0, 1 and 2, with osteoblast markers increased in the ablated cells in all clusters. It is not clear why this does not drive the cells to cluster separately per condition. It would be of interest to compare the expression levels of these markers between the clusters to understand how the expression levels of these markers compare between all of the clusters/conditions.

<Response>

We would like to point out that the number of genes differentially expressed between two conditions (CONT and ABL) was less than 500 genes among ~4,000 genes expressed by cells. Expression levels of the representative cell type specific marker genes such as *Cxcl12* and *Adipoq* were unchanged between two conditions. Therefore, we assume that these cells did not lose their identities as reticular cells despite some upregulation of osteoblast-signature genes, not driving these cells cluster separately.

To meet the reviewer's request, we have demonstrated violin plots comparing the expression levels of representative marker genes among the clusters, split by conditions (e.g. CONT vs. ABL) for each biological replicate. We can appreciate that there are some differences in terms of expression osteoblast-signature genes such as *Postn*, *Spp1* and *Runx2* between two conditions.

Figure R3-4

10. <Reviewer>

The third set of single cell data is another ablation experiment in which again tdTomato cells were sorted and analysed by 10x genomics. Again, only a fraction of the cells (1825/9474 cells) expressed the tdTomato transcript. These cells show similar clustering to the previous experiment (here there is not control femur). Pseudotime ordering of these cells was used to confirm a linear "trajectory" from reticular cell to pre-

osteoblast to osteoblast, and that Wnt signalling components displayed regulated expression along this trajectory.

<Response>

Following the reviewer's suggestion, we have clarified this issue regarding contamination of tdTomato-negative cells in the revised manuscript.

The following sentences were added to the Results, under the first section:

Initial analysis revealed again that a substantial fraction of cells (7,074/9,469 cells) belonged to the clusters without detectable tdTomato expression.

11. <Reviewer>

Overall I think the single cell approaches used in the paper are appropriate, and serve the purpose of hypothesis generation in the manuscript. However the authors are drawing conclusions from the single-cell data that are beyond the technical limitations of the approach – from single replicate experiments it is simply not possible to quantify reliably changes in cell abundance and even measurements of differential expression between samples can be confounded by batch effects. I am also not clear on why so few of the sorted cells express the selection marker (perhaps I have missed something) and why there is no discussion of the 'other cells' that make up most of the single-cell data in the paper. As I said, this is right at the technical limits of what 10X genomics can do in terms of gene detection and cell counting, and the authors should critically appraise this in any revision of the manuscript.

<Response>

Thank you very much for these extremely important comments.

First, we have performed additional biological replicate runs of single cell RNA-seq experiments particularly in Figure 1 (two independent mice) and Figure 3 (two independent groups of mice), where changes in cell abundance and/or differential expression of between groups are inferred, in the revised manuscript.

Second, we have added technical details and more information on FACS-sorting of fluorescently labeled bone marrow stromal cells, along with additional descriptions on contamination of hematopoietic cells in the two-dimensional plots in the revised manuscript.

We have changed the wording in the revised manuscript to acknowledge the technical limitations associated with droplet-based single RNA-seq experiments.

12. <Reviewer>

Bone marrow stromal cells – CXCL12+ (cre-ER mice)

Injury makes CXcl12+ cells convert identity into skeletal stem cell-like state and upregulation of osteoblast genes including Wnt. B-catenin down = drop in bone marrow regeneration. MEDIATED BY CANONICAL WNT?

tG characterisation -not mature osteoblasts (Col1a1 negative) subset of LepR+ quiescent single cell Cxcl12-GFP+ cells were heterogeneous and clustered into four groups, including two clusters of mesenchymal (Cluster 0,1 – pre osteoblast and adipocyte) and endothelial (Cluster 2,3) Cluster 1 enriched for tdTomato – a dozen or so cells in how many? Would have been better to sort tdTomato Cluster 0 were more proliferative – what does this mean in the context of a static measurement. Few tomato + cells in cluster 1 Tomato cells are rich in adipocyte related genes and cytokines. In vitro give adipocytes chondrocytes and osteoblasts. Can self renew
Quantitative nature of sc-RNA for cell population counts, appears to be a single experiment with no replication and a very small number of cells in the population enriched for tdTomato
Ablated/control Saying something is not expressed is not possible with 10x data – was not detected.

1. Comments for Author

What are the major claims of the paper?

Are they novel and will they be of interest to others in the community and the wider field?

If the conclusions are not original, it would be helpful if you could provide relevant references. Is the work convincing, and if not, what further evidence would be required to strengthen the conclusions?

On a more subjective note, do you feel that the paper will influence thinking in the field? Please feel free to raise any further questions and concerns about the paper.

We would also be grateful if you could comment on the appropriateness and validity of any statistical analysis, as well the ability of a researcher to reproduce the work, given the level of detail provided.

To increase the transparency and openness of the reviewing process, we do support our reviewers signing their reports to authors if the reviewers feel comfortable doing so. If, however, you prefer to send an anonymized report we will continue to respect and maintain your anonymity. Referee reports, whether signed or not, are subsequently shared with the other reviewers.

<Response>

We assume that these write-ups are a carryover from the reviewer's note, which was accidentally copied and pasted into the 'comments to the authors' section.

REVIEWERS' COMMENTS:

Reviewer #1 (Remarks to the Author):

This is a very good paper, with important implications for the field.

Bo and Sean

Reviewer #2 (Remarks to the Author):

This manuscript is significantly improved. There is one area that should be clarified. Beta-catenin has different effects at different protein levels, and APC regulation of beta-catenin will provide a different level of stabilization than removing the phosphorylation sites encoded by exon three. This should be noted as a limitation of the approach of stabilizing beta-catenin by targeting APC.

Reviewer #3 (Remarks to the Author):

The authors have, for the most part, addressed my concerns about the initial submission.

The contamination of the tdTomato / GFP cells with hematopoietic cells remains something of a concern; perhaps some clarification would help - the authors "hypothesize that these cellular doublets become dissociated when they are loaded onto the oil droplet in the 10X Genomics 3' assay chip" - this is quite speculative.

I would like to understand better why the authors did not use a CD45 negative population to capture the cells of interest (as in figure 1F, FACS analysis) in a more pure format?

Also, could the authors show a plot of ptpcr/CD45 expression in the tdTomato positive (by 10x cells)? I would just like to understand if this contamination carries over into the 10x data.

This is really my only outstanding concern - I would like to understand better the effect of these cellular intertwinings on the data presented

1. <Reviewer #1>

This is a very good paper, with important implications for the field.

<Response>

We are very excited that our paper is coming out soon!

2. <Reviewer #2>

This manuscript is significantly improved. There is one area that should be clarified. Beta-catenin has different effects at different protein levels, and APC regulation of beta-catenin will provide a different level of stabilization than removing the phosphorylation sites encoded by exon three. This should be noted as a limitation of the approach of stabilizing beta-catenin by targeting APC.

<Response>

We would appreciate your very important comments. Following the reviewer's suggestion, we have now added the following sentence to clarify the limitation of our APC experiments.

The following underlined clauses were added to the Results, under the last section:

Therefore, a high level of β -catenin in Cxcl12-creER⁺ BMSCs due to APC haploinsufficiency does not inhibit their osteoblast differentiation during cortical bone regeneration. A limitation of this approach is that it provides a different level of β -catenin stabilization than removing the phosphorylation sites encoded by exon 3.

3. <Reviewer #3>

The authors have, for the most part, addressed my concerns about the initial submission. The contamination of the tdTomato / GFP cells with hematopoietic cells remains something of a concern; perhaps some clarification would help - the authors "hypothesize that these cellular doublets become dissociated when they are loaded onto the oil droplet in the 10X Genomics 3' assay chip" - this is quite speculative. I would like to understand better why the authors did not use a CD45 negative population to capture the cells of interest (as in figure 1F, FACS analysis) in a more pure format?

<Response>

We completely agree with the reviewer that the issue related to hematopoietic cell contamination remains a concern. We have cited three recent papers on bone marrow stromal cell datasets showing some degree of hematopoietic cell contamination in the revised manuscript.

We decided not to use CD45 antibodies to negatively select hematopoietic cells, because we aimed to minimize the time between cell dissociation and cell sorting, in order to minimize a common issue in skewed transcriptomes due to long handling time.

We would like to address this fundamentally important topic in our future endeavor.

4. <Reviewer #3>

Also, could the authors show a plot of *ptprc*/CD45 expression in the *tdTomato* positive (by 10x cells)? I would just like to understand if this contamination carries over into the 10x data. This is really my only outstanding concern - I would like to understand better the effect of these cellular intertwinings on the data presented.

<Response>

Thank you very much for your additional suggestion. Below, we have shown the feature plots for *Ptprc* (encoding CD45) and *tdTomato-WPRE* (encoding *tdTomato*) for the single cell RNA-seq data shown on the revised **Figure 2**. Cells expressing *tdTomato* do not appear to express *Ptprc*, at least based on these plots. It is very interesting that contamination of hematopoietic cells at the stage of FACS sorting does not carry over into the 10X scRNA-seq data.

This data has been incorporated as part of the revised **Supplementary Figure 2**.

Figure R3-1 (part of the revised Supplementary Figure 2)